# Objective Soups: Multilingual Multi-Task Acoustic Modeling for Speech Processing

## Abstract

The need for training multilingual multi-task automatic speech recognition (ASR) models is increasingly evident. However, a significant challenge arises from the conflicts among multiple objectives when using a single model. Multi-objective optimization (MOO) can address this challenge by facilitating the optimization of multiple conflicting objectives and aligning the gradient updates in a common descent direction. While MOO helps avoid conflicting gradient update directions, a critical issue is that when there are many objectives such as those in multilingual multi-task ASR, it is often *impossible to find* such common descent directions. Therefore, an interesting question is: would it be more effective to separate highly conflicting objectives into different optimization levels or keep them in one level? To address this question, this paper investigates three multi-objective ASR training formulations, which we refer to as **objective soup recipes**. These formulations use MOO at different optimization levels to mitigate potential conflicts among all objectives. We conduct an extensive investigation using the LibriSpeech and AISHELL v1 datasets for ASR, along with the CoVoST v2 dataset for both ASR and speech-to-text translation (S2TT) tasks, to determine the highly conflicting objectives and the optimal training recipes among these three MOO training algorithms.

## 1 Introduction

Automatic Speech Recognition (ASR) technology is crucial across various applications such as virtual assistants and voice search (Graves et al., 2013b; Hinton et al., 2012). Its importance extends to multilingual environments, where there is a growing demand for ASR systems capable of processing multiple languages efficiently. Multilingual ASR systems find use in international communication and language learning platforms (Toshniwal et al., 2018; Yadav & Sitaram, 2022). Ideally, those multilingual systems can perform multiple tasks like transcription and translation simultaneously (Chen & Mak, 2015). Unified speech models that handle multiple tasks across diverse languages have emerged as a promising solution (Schultz & Kirchhoff, 2006; Bourlard et al., 2011), simplifying maintenance efforts and reducing system complexity. However, training a unified model for multilingual multi-task learning is challenging due to language diversity, task heterogeneity, data scarcity, and model complexity (Kim et al., 2021; Fu et al., 2022).

A common approach to tackling these challenges is to introduce different objective functions that represent different performance metrics and integrate them into the ASR training process. For example, to overcome data scarcity, one can introduce both the self-supervised learning (SSL) loss and the supervised learning loss in ASR tasks (Oord et al., 2018; Schneider et al., 2019; Baevski et al., 2019; 2020; Hsu et al., 2021); to address multilingual phonetic diversity, one can introduce separate objective functions for each language and also enforce fairness across languages. We call this methodology as **objective soup**. In this context, multilingual multi-task learning naturally presents a multi-objective learning problem, but the caveat is that different objectives may conflict with each other – improvement of some objectives degrades others. Notably, a related work to ours is *rewarded soups* (Rame et al., 2024), where the goal is to achieve the Pareto-optimal alignment for foundation models by a weighted combination of diverse reward model parameters. The resulting reward model provides an objective for alignment. Different from this, we consider a broader range of MOO methods to tackle objective conflicts.

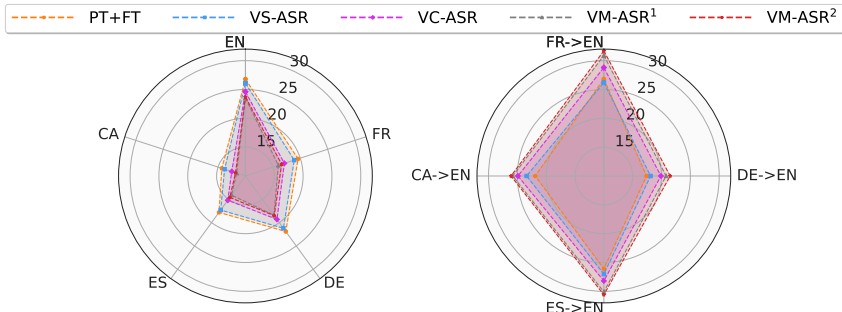

Figure 1: Radar plots of ASR (left, WER) and S2TT (right, BLEU score) performance by optimization technique for a 100M parameter model. **Closer proximity to the origin** indicates better ASR performance, while **greater distance** indicates better S2TT performance.

To handle conflicts among multiple objectives, recent studies have leveraged multi-objective optimization (MOO) to tackle multilingual multi-tasking ASR problems. Roughly speaking, there are three primary MOO formulations, specified in Section 3, to model multilingual multi-tasking ASR problems: i) the single-level vector optimization method; ii) the bilevel hybrid vector optimization method; and, iii) the multi-level optimization (MLO) method (Miettinen, 1999). However, which method is most suitable for multilingual multi-task ASR remains unclear, especially when the objectives exhibit conflict. Selecting an optimal solution from this front involves making challenging decisions regarding acceptable trade-offs, which can negatively impact the model's performance. Therefore, identifying the most appropriate MOO-based algorithm for multilingual multi-task ASR remains a significant research challenge.

In this context, we aim to thoroughly investigate these three MOO-based algorithms for multilingual multi-task acoustic modeling and determine their effectiveness in handling higher conflicting objectives. By evaluating their performance, we hope to identify the algorithm that best balances the trade-offs and enhances the overall model performance.

**Our findings and contributions.** We conduct extensive experiments on various widely-used benchmark datasets, including LibriSpeech, AISHELL, and CoVoST v2, and across models with different sizes. We find consistent performance gains through MLO of self-supervised and supervised objectives for ASR and S2TT tasks across multiple languages. The findings are summarized below.

- **F1: MOO methods mitigate gradient conflicts in pre-training (PT) and fine-tuning (FT), thus improving the performance.** Compared to traditional PT+FT methods that are either implemented in a two-stage manner or through static weighting, the MOO method with dynamic weighting to handle conflicting gradients performs better. This is because MOO methods mitigate conflicting multilingual multitask objectives through optimization along common descent directions. On average, the use of MOO (VC-ASR[1]) improves the ASR and S2TT performance over Joint PT+FT without MOO by 3.8% and 4.8%, respectively. See results in Tables 1 and 2.

- **F2: Hierarchical objectives enhance ASR performance.** Introducing appropriate hierarchy in multilingual multi-task ASR objectives consistently improves ASR performance. In particular, the MLO method consistently outperforms both single-level and bilevel optimization methods. This suggests that separating highly conflicting objectives across multiple optimization levels effectively mitigates conflicts. On average, MLO (VM-ASR[2]) improves ASR and S2TT performance by 5.6% and 5.9%, respectively, compared to VC-ASR. Refer to Tables 1 and 2 for details.

- **F3: Task-based hierarchy outperforms language-based hierarchy in both efficiency and accuracy.** In MLO, a task-based hierarchy requires fewer levels compared to a language-based hierarchy, thereby reducing the overall complexity of the optimization algorithms. Moreover, the task-based hierarchy achieves superior accuracy, as task-related objective conflicts tend to be more significant than language-related objective conflicts. Refer to Figure 3 for an illustration of gradient conflicts.

---

[1]Vectorized objectives with lower-level constraint for ASR

[2]Vectorized multilevel ASR

**F4: The penalty parameter used in the multilevel reformulation plays a crucial role.** Our studies reveal that, while large penalty parameters used in the reformulation of multilevel speech optimization theoretically guarantee good convergence of lower-level objectives, they may adversely affect the generalization performance of the learned ASR model. Well-calibrated penalty parameters, however, can improve overall ASR and S2TT performance by 8.3% and 2.2%, respectively. See the results in Tables 3 and 4.

## 2 RELATED WORK

In this section, we review existing works on multilingual ASR and S2TT, aiming to identify the current research landscape in these areas.

**Multilingual ASR and S2TT.** Earlier works in multilingual ASR used deep neural networks, hidden Markov models, and multilayer perceptron models (Heigold et al., 2013; Thomas et al., 2010; Tüske et al., 2013; Ghoshal et al., 2013). Later studies showed Long Short-Term Memory (LSTM) models to be more effective for multilingual ASR (Graves et al., 2013a; Zhou et al., 2017). Recently, Seq2Seq models with hybrid attention/CTC algorithms and transformer-based models have achieved state-of-the-art results (Watanabe et al., 2017; Toshniwal et al., 2018; Zhou et al., 2018). Multilingual S2TT tasks have also gained attention, primarily using transformer-based models with SSL pre-training (Li et al., 2020; Bapna et al., 2022; Ren et al., 2020). Despite advancements, these systems lack multi-tasking capabilities, a longstanding challenge in developing a single model for multiple speech-related tasks. This line of work is orthogonal to the current paper and can potentially be combined with our multi-objective training recipes.

**Multi-task learning for speech recognition.** Multi-task learning for joint ASR and S2TT tasks has been explored in various studies, yet challenges remain in optimizing shared representations and reducing task interference. The first algorithm for joint ASR and S2TT decoding was introduced by (Anastasopoulos & Chiang, 2018). Subsequent models improved this by using word embedding intermediates and two-stage models (Chuang et al., 2020; Sperber et al., 2019). A transformer-based dual encoder-decoder architecture with separate decoders for each task was also applied (Le et al., 2020). The Whisper (Radford et al., 2023) model was trained on large-scale audio dataset for multitask learning. The Mu$^2$SLAM model (Cheng et al., 2023) pre-trains on multilingual speech, text, and supervised data. Cross-modality learning from multiple self-supervised and supervised subtasks establishes a robust multi-task algorithm (Tang et al., 2022). Joint pre-training and fine-tuning is also explored in ASR and multilingual multitask speech-to-text tasks to reduce training complexity (Bai et al., 2022; Saif et al., 2024; Talnikar et al., 2021). Although these approaches address multilingual multi-task learning using static weighting or constrained optimization, they do not explicitly tackle conflicting objectives such as using a conflict-avoidant update direction, which may lead to suboptimal results.

In this paper, we investigate conflicting objectives in multilingual multitask speech-to-text tasks and propose MOO-based algorithms to mitigate these conflicts. Our approach demonstrates a significant improvement over baseline methods, highlighting the effectiveness of MOO in multilingual multitask speech-to-text tasks.

## 3 UNIFYING MOO TRAINING METHODS

In this section, we introduce multi-objective optimization and its optimality condition, discuss three potential problem formulations, and present the corresponding algorithms to solve these problems.

### 3.1 MULTI-OBJECTIVE OPTIMIZATION: A PRIMER

The goal of MOO is to learn a model that simultaneously optimizes multiple objectives, where objectives can represent different tasks or learning metrics. Let $\Theta \in \mathbb{R}^q$ denote the model parameter. Given $M$ objectives, each denoted as $l_m(\Theta)$, for $m \in [M]$, the general MOO problem solves

$$\min_{\Theta \in \mathbb{R}^q} \quad L(\Theta) := [l_1(\Theta), \ldots, l_M(\Theta)]. \tag{1}$$

We use the following necessary optimality condition for MOO.

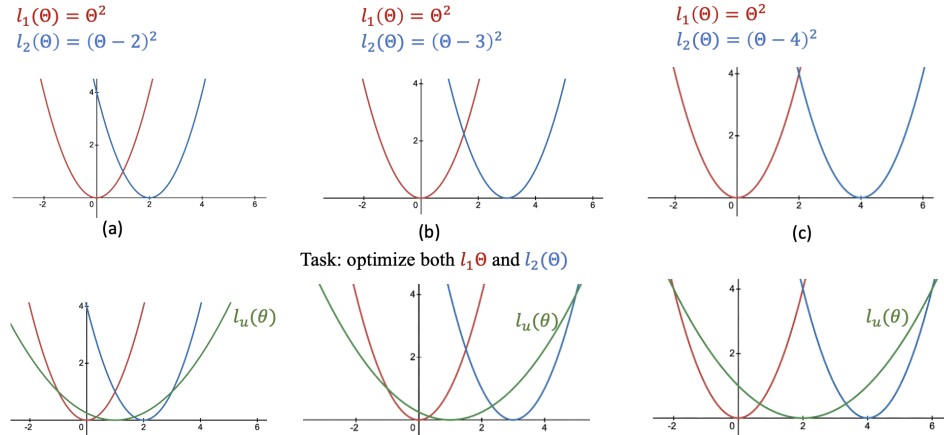

Figure 2: Intensified conflicts among optimization objectives expand the search region for Pareto optimal points, challenging algorithms. Using a lower-level constraint $l_u(\theta)$, such as the self-supervised loss in ASR tasks, effectively reduces this region, simplifying the algorithm's task.

**Definition 1** (Pareto stationary). A model $\Theta$ is Pareto stationary if there exists $\lambda \in \Delta^M := \{\lambda \in \mathbb{R}^\top \mid \mathbf{1}^\top \lambda = 1,\ \lambda \geq 0\}$ such that $\nabla L(\Theta)\lambda = 0$, i.e., $\min_{\lambda \in \Delta^M} \|\nabla L(\Theta)\lambda\| = 0$.

### 3.2 THREE MOO FORMULATIONS OF MULTILINGUAL MULTI-TASK ASR

We adopt a joint PT+FT training approach for our MOO-based multilingual multitask speech-to-text algorithms, facilitating the sequential optimization of PT and FT objectives. This results in locally matched optima that enhance model convergence and overall performance (Saif et al., 2024). Unlike (Saif et al., 2024), which employs SSL objective as a lower-level constraint to establish a feedback loop between PT and FT, we leverage SSL objective to narrow the search space for identifying the most suitable Pareto optimal point in the Pareto optimal front. Additionally, we implement MOO to address conflicts between objectives, an aspect not explored in their work.

In our formulation, let $\Theta := [\theta; \phi]$, where $\theta$ is the parameter of the backbone and $\phi$ is that of a language/task-dependent layer of a model. For pre-training shared backbone parameters $\theta$, we use SSL loss, $l_u(\theta)$. For language and task specific parameters $\phi_{t,n}$, we use supervised classification loss, $l_{ctc}(\theta, \phi_{t,n})$, where $t \in [T]$ and $n \in [N]$ represents different languages and tasks, respectively. This self-supervised loss and multiple supervised losses form the MOO objectives for multilingual ASR. Note that, for multi-objective ASR, we can represent all the objectives as a vector, $L(\Theta)$ containing supervised losses from different languages and tasks such as multilingual ASR and S2TT where $\Theta := [\theta, \phi_{1,1}, \cdots, \phi_{T,N}]$; that is, $L(\Theta) := [l_{ctc}(\theta, \phi_{1,1}), \ldots, l_{ctc}(\theta, \phi_{T,N})]$.

Our final goal is to learn a multilingual multi-task model with a shared backbone parameterized by $\theta$, and a task and language-specific part, each parameterized by $\phi_{t,n}$, $\forall t \in [T], \forall n \in [N]$. To learn these parameters while avoiding conflicting gradient directions we formulate three MOO ASR problems. We discuss these formulations below:

**Vectorized single-level ASR (VS-ASR).** In this formulation, we treat all the objectives as single-level vectorized objectives without any lower-level constraints. Hence, the problem formulation is,

$$\min_{\Theta \in \mathbb{R}^q} \left[ \underbrace{l_{ctc}(\theta, \phi_{1,1}), \cdots, l_{ctc}(\theta, \phi_{1,N})}_{\text{1-st language with } N \text{ tasks}}, \ldots, \underbrace{l_{ctc}(\theta, \phi_{T,1}), \cdots, l_{ctc}(\theta, \phi_{T,N})}_{T\text{-th language with } N \text{ tasks}}, l_u(\theta) \right]. \quad (2)$$

**Vectorized objectives with lower-level constraint for ASR (VC-ASR).** To mitigate the challenge of conflicting objectives and reduce the search space for an optimal Pareto stationary point, incorporating a suitable lower-level constraint, $l_u(\theta)$, can be beneficial (see Figure 2) (Miettinen, 1999). However, $l_u(\theta)$ must possess certain essential properties. Its gradient update direction should have minimal conflict with the gradient directions of other objectives, as increased conflict would hinder its role in narrowing the search space for the common optimal point. Moreover, the optimization space defined by this constraint must be sufficiently flat, ensuring that the common optimal point across all objectives lies within it. In this context, we incorporate the self-supervised loss as a lower-level

constraint, as it exhibits these desirable properties (see Appendix C). This approach helps align the gradient directions and maintain a feasible optimization region, ultimately enhancing overall performance. By constraining the self-supervised loss to be smaller than a threshold $\epsilon$, our VC-ASR method can be formulated as

$$\min_{\Theta \in \mathbb{R}^q} \left[ l_{\text{ctc}}(\theta, \phi_{1,1}), \cdots, l_{\text{ctc}}(\theta, \phi_{1,N}), \ldots, l_{\text{ctc}}(\theta, \phi_{T,1}), \cdots, l_{\text{ctc}}(\theta, \phi_{T,N}) \right]$$

$$\text{s.t. } l_{\text{u}}(\theta) - \min_{\theta} l_{\text{u}}(\theta) \leq \epsilon. \tag{3}$$

This formulation aims to minimize the vector of the supervised losses, $L(\Theta)$, subject to a constraint that another self-supervised loss function, $l_{\text{u}}(\theta) - \min_{\theta} l_{\text{u}}(\theta)$, remains below a specified threshold $\epsilon$. Consequently, this $\epsilon$ constraint defines the feasible region for the upper-level objectives, ensuring the attainment of a Pareto stationary point. Our investigation has revealed that employing the self-supervised objective as a lower-level constraint for ASR tasks yields optimal results. This observation validates the algorithm of our VC-ASR, where we separate the self-supervised objective from the supervised objectives and optimize them at the lower and upper levels, respectively. Additionally, this formulation facilitates joint lower and upper-level training, enhancing the overall optimization process.

**Vectorized multilevel ASR (VM-ASR).** Building upon the VC-ASR formulation, we introduce VM-ASR, a multilevel multilingual multi-task ASR algorithm. Through VM-ASR, we aim to explore whether extending our VC-ASR algorithm into a MLO framework based on tasks and languages offers advantages and mitigates the risk of being trapped in sub-optimal Pareto stationary points. In MLO, decision-making follows a hierarchical structure, with decisions made at different levels within the hierarchy. The problem formulation for multilevel ASR optimization can be expressed as follows:

$$\operatorname*{argmin}_{\phi_1 \in \mathbb{R}^r, \phi_2^*, \ldots, \theta^*} L_1(\phi_1, \phi_2^*, \ldots, \theta^*) \ldots$$

$$\text{s.t. } \phi_p^* = \operatorname*{argmin}_{\phi_p \in \mathbb{R}^r, \phi_{p+1}^*, \ldots, \theta^*} L_p(\phi_1, \cdots, \phi_p, \phi_{p+1}^*, \ldots, \theta^*) \ldots \tag{4}$$

$$\text{s.t. } \theta^* = \operatorname*{argmin}_{\theta \in \mathbb{R}^s} L_P(\phi_1, \phi_2, \ldots, \theta),$$

where $L_p$ is a vector of ASR objectives and $\forall p \in [P]$ is the optimization level. In VM-ASR, training is performed on multiple levels, with feedback across different levels. We employ separate classification heads for each task. They share a backbone encoder layer. it follows that the optimization of all task-specific parameters, denoted as $\phi_{t,n}$, is contingent upon optimizing the backbone parameters, $\theta$. Consequently, the self-supervised objective is placed at the lowest level of the optimization hierarchy.

# 4 APPLICATIONS OF VS-ASR, VC-ASR, AND VM-ASR

In this section, we evaluate the three multi-objective formulations introduced in Section 3 on the multilingual multi-task ASR problem, including ASR and S2TT tasks.

## 4.1 DEFINING OBJECTIVES

**Objectives of self-supervised and supervised training.** For SSL and supervised learning we use Contrastive Predictive Coding (CPC)[3] loss (Oord et al., 2018), $l_{\text{u}}(\theta)$ and Connectionist Temporal Classification (CTC) loss (Graves et al., 2006), $l_{\text{ctc}}(\theta, \phi)$, respectively. Here, $\theta$ represents the backbone parameters, shared by all the objectives and $\phi$ is the parameters of the task-specific classification heads. We formulate the joint SSL and supervised learning as a MOO problem and solve it using VS-ASR, VC-ASR, and VM-ASR techniques.

**Objectives of language-specific outputs.** We consider the same loss function on different languages as distinct objectives, each with its own classification heads and classification loss, denoted as $l_{\text{ctc}}(\theta, \phi_t)$, where $t \in [T]$ represents a specific language.

**Objectives of ASR and S2TT.** Objectives of ASR and S2TT tasks are considered to be distinct objectives. We use two different classification heads for ASR and S2TT. We use $l_{\text{ctc}}(\theta, \phi_{t,1})$ and

---

[3]Training results using the more advanced pre-training methods, BEST-RQ and Wav2Vec2, are presented in Appendix F, in Tables 7 and 8.

$l_{\text{ctc}}(\theta, \phi_{t,2})$ losses for ASR and S2TT tasks, respectively, where $\phi_{t,1}$ and $\phi_{t,2}$ represent the parameters of the classification heads for the ASR and S2TT tasks, respectively.

Given this problem, we formulate multilingual VS-ASR, VC-ASR, and VM-ASR for ASR and S2TT. **Remark** 1. For MLO, objectives are prioritized based on their importance. In VM-ASR, this includes task-based and language-based MLO. Task-based MLO experiments with ASR and S2TT, alternating their primary and secondary levels. Language-based MLO involves English (LibriSpeech) and Chinese (AISHELL), also alternating their primary and secondary levels.

To update the backbone parameters, $\theta$, and task-specific parameters, $\phi$, in the three algorithms, we use a gradient-based algorithm. Detailed descriptions and derivations of these algorithms are provided in Appendix A. Below we provide a task-specific update rule for ASR and S2TT tasks.

## 4.2 VECTORIZED SINGLE-LEVEL ASR (VS-ASR)

For single vectorized objective training, we can optimize the vectorized objectives using Algorithm 1 in Appendix A where the shared backbone parameters are updated using

$$\theta^{k+1} = \theta^k - \alpha \sum_{t=1}^{T} \lambda_{t,1}^k \nabla_\theta l_{\text{ctc}}(\theta^k, \phi_{t,1}^k) - \alpha \sum_{t=1}^{T} \lambda_{t,2}^k \nabla_\theta l_{\text{ctc}}(\theta^k, \phi_{t,2}^k) - \alpha \lambda_u^k \nabla_\theta l_{\text{u}}(\theta^k). \quad (5)$$

In this context, $\alpha > 0$ is the learning rate assigned to the backbone parameters. Moreover, $\lambda_{t,1}^k$ and $\lambda_{t,2}^k$ represent the dynamic update directions for ASR and S2TT objectives, respectively, which are computed using the MoDo algorithm (Chen et al., 2023). Here, $\lambda_u^k$ is the dynamic update directions for self-supervised objective, $l_{\text{u}}$. Similarly, taking the gradients of each of the supervised objective functions with respect to task-specific output heads, the task-specific output parameters are updated via

$$\phi_{t,1}^{k+1} = \phi_{t,1}^k - \beta \nabla_\phi l_{\text{ctc}}(\phi_{t,1}^k, \theta^k) \quad \text{and} \quad \phi_{t,2}^{k+1} = \phi_{t,2}^k - \beta \nabla_\phi l_{\text{ctc}}(\phi_{t,2}^k, \theta^k) \quad (6)$$

where $\beta > 0$ is the learning rate for the task-specific parameter.

## 4.3 VECTORIZED OBJECTIVES WITH LOWER-LEVEL CONSTRAINT FOR ASR (VC-ASR)

To train a model using the VC-ASR algorithm, the backbone parameters $\theta$ are updated using

$$\theta^{k+1} = \theta^k - \alpha \sum_{t=1}^{T} \lambda_{t,1}^k \nabla_\theta l_{\text{ctc}}(\theta^k, \phi_{t,1}^k) - \alpha \sum_{t=1}^{T} \lambda_{t,2}^k \nabla_\theta l_{\text{ctc}}(\theta^k, \phi_{t,2}^k) - \alpha \eta \nabla_\theta l_{\text{u}}(\theta^k). \quad (7)$$

To update task-specific heads, we employ (6); see a summary in Algorithm 2.

## 4.4 VECTORIZED MULTILEVEL ASR (VM-ASR)

In VM-ASR, we update the backbone parameters $\theta$ using the following equation

$$\theta^{k+1} = \theta^k - \alpha \left( \sum_{t=1}^{T} \lambda_{t,1}^k \nabla_\theta l_{\text{ctc}}(\theta^k, \phi_{t,1}^k) + \eta_1 \sum_{t=1}^{T} \lambda_{t,2}^k \nabla_\theta l_{\text{ctc}}(\theta^k, \phi_{t,2}^k) + \eta \nabla_\theta l_{\text{u}}(\theta^k) \right) \quad (8)$$

where $\eta_1$ and $\eta$ are penalty parameters. We update task-specific classification parameters using (6); see a summary in Algorithm 3.

## 5 EXPERIMENTAL RESULTS AND FINDINGS

In this section, we conduct numerical simulations for proposed three algorithms, two-stage PT+FT, static weighting (Gong et al., 2022), parameter efficient fine-tuning (PEFT) and joint PT+FT without (W/O) MOO (Saif et al., 2024)[4], to determine the optimal MOO ASR algorithm when optimizing

---

[4]Description of these methods is added in Appendix: D

Table 1: ASR WERs comparison using CoVoST 2, including PT+FT, Two-stage static, Joint PT+FT without (W/O) MOO, PEFT, VS-ASR, VC-ASR, and VM-ASR with different model sizes (100M and 58M). VM-ASR optimizes objectives using UAS (self-supervised → ASR → S2TT) and USA (self-supervised → S2TT → ASR) sequences.

| Param | Lang | Two-stage (PT+FT) | Two-stage static | Joint PT+FT W/O MOO | PEFT | VS-ASR | VC-ASR | VM-ASR UAS | VM-ASR USA |
|---|---|---|---|---|---|---|---|---|---|
| 100M | En | 26.8% | 27.3% | 25.2% | 27.9% | 26.1% | 24.6% | **23.5%** | 23.7% |
| | Fr | 19.6% | 19.4% | 17.8% | 21.5% | 18.9% | 17.1% | **16.0%** | 16.6% |
| | De | 21.9% | 21.8% | 20.2% | 23.8% | 21.2% | 19.3% | **18.4%** | 18.5% |
| | Es | 17.8% | 17.2% | 15.9% | 19.6% | 17.3% | 15.2% | **14.1%** | 14.6% |
| | Ca | 14.3% | 13.7% | 13.1% | 16.7% | 13.8% | 12.5% | **11.6%** | 11.8% |
| | Ave. | 20.1% | 19.9% | 18.4% | 21.9% | 18.8% | 17.7% | **16.7%** | 17.0% |
| 58M | En | 29.7% | 29.8% | 28.4% | 30.2% | 29.2% | 27.9% | **26.8%** | 27.1% |
| | Fr | 26.5% | 26.4% | 25.9% | 28.2% | 26.1% | 25.2% | **24.3%** | 24.7% |
| | De | 28.8% | 28.6% | 27.8% | 30.1% | 28.3% | 27.1% | **26.2%** | 26.8% |
| | Es | 21.3% | 21.2% | 20.4% | 22.3% | 20.9% | 19.4% | **18.9%** | 19.1% |
| | Ca | 18.2% | 17.9% | 17.5% | 18.8% | 18.0% | 16.9% | **16.2%** | 16.5% |
| | Ave. | 24.9% | 24.8% | 24.0% | 25.9% | 24.5% | 23.3% | **22.1%** | 22.8% |

conflicting objectives. This is crucial to avoid the risk of the model getting stuck in a suboptimal Pareto stationary point. We analyzed ASR and S2TT performance in a multilingual multi-task setup using the CoVoST 2 dataset, selecting five languages for ASR (En, Fr, De, Es, Ca)[5] and four for S2TT (Fr, De, Es, Ca). Additionally, we performed experiments with a combination of the LibriSpeech and AISHELL datasets. Our results demonstrate that the MOO approaches consistently outperforms the joint PT+FT W/O MOO method and other baselines, confirming their effectiveness in achieving better ASR and S2TT performance.

**Models and hyper-parameters:** We use two Conformer models (Gulati et al., 2020) for multilingual multi-task ASR experiments. The first model has 10 Conformer blocks with a hidden dimension of 612 units and 12 attention heads; the second model has 8 blocks with 512 hidden dimensions and 8 attention heads. Each attention head has a dimension of 51 for the first model and 64 for the second model. Both configurations use a convolutional kernel size of 31 to capture temporal dependencies and distinct classification heads with varying output sizes. For VC-ASR, the initial penalty parameter $\eta$ is 0, increasing by 0.02 per epoch. For VM-ASR, there are three optimization levels. The penalty constant $\eta_1$ used for the second level starts at 0.1, increasing by 0.02 per epoch, while the lower-level penalty constant $\eta_2$ starts at 0 and increases by 0.02 per epoch. We use learning rates of $\alpha = 5 \times 10^{-4}$ for backbone parameters and $\beta = 5 \times 10^{-5}$ for classification parameters.

**Training time and memory complexity:** Our proposed MOO ASR models exhibit higher training time and memory complexity compared to traditional PT+FT models. Specifically, the PT+FT model uses an average of 8.7 GB of GPU memory and takes approximately 2.25 hours per epoch, while the MOO ASR models require about 11.6 GB of GPU memory and around 2.8 hours per epoch. This increased memory consumption and training time are primarily due to the additional gradient calculations needed for computing dynamic weights. However, the higher training cost is justified by significant performance gains. Moreover, our MOO approach demonstrates scalability as the number of tasks increases, and in the long run, a single MOO model reduces resource demands during deployment, making it a more efficient solution overall G.

Based on our experiments, we summarize the following observations[6]:

## 5.1 ENHANCING ASR AND S2TT PERFORMANCE WITH MLO

"Multilevel optimization significantly improves ASR and S2TT performance by effectively balancing learning objectives and narrowing the search for optimal Pareto stationary points."

---

[5]English (En), French (Fr), German (De), Spanish (Es), Catalan (Ca)
[6]Additional result tables and discussion can be found in Appendix: F.

Table 2: S2TT average(Ave.) BLEU score comparison using the CoVoST 2 dataset, including PT+FT, Two-stage static, Joint PT+FT without(W/O) MOO, PEFT, VS-ASR, VC-ASR, and VM-ASR with different parameter sizes (100M and 58M). VM-ASR optimizes objectives using UAS (self-supervised $\rightarrow$ ASR $\rightarrow$ S2TT) and USA (self-supervised $\rightarrow$ S2TT $\rightarrow$ ASR) sequences.

| Param | Lang→Eng | Two-stage (PT+FT) | Two-stage static | Joint PT+FT W/O MOO | PEFT | VS-ASR | VC-ASR | VM-ASR UAS | VM-ASR USA |
|---|---|---|---|---|---|---|---|---|---|
| | Fr→En | 26.8 | 26.8 | 27.4 | 25.3 | 26.2 | 28.8 | 30.9 | **31.7** |
| | De→En | 17.4 | 17.5 | 18.9 | 15.9 | 18.1 | 19.9 | 20.8 | **21.5** |
| | Es→En | 26.1 | 26.3 | 27.3 | 24.7 | 27.0 | 28.2 | 30.1 | **30.6** |
| 100M | Ca→En | 21.9 | 22.0 | 23.5 | 20.2 | 23.4 | 24.9 | 25.8 | **26.1** |
| | Ave. | 23.0 | 23.1 | 24.3 | 21.5 | 23.7 | 25.4 | 26.9 | **27.5** |
| | Fr→En | 23.4 | 23.5 | 24.1 | 22.2 | 23.9 | 25.8 | 26.5 | **26.8** |
| | De→En | 15.0 | 15.1 | 15.4 | 13.6 | 15.3 | 16.2 | 17.1 | **17.4** |
| | Es→En | 24.0 | 24.2 | 24.4 | 22.5 | 24.2 | 25.1 | 25.6 | **25.9** |
| 58M | Ca→En | 19.4 | 19.2 | 19.2 | 17.9 | 19.1 | 20.3 | 21.4 | **21.6** |
| | Ave. | 20.4 | 20.5 | 20.8 | 19.0 | 20.6 | 21.8 | 22.6 | **22.9** |

Table 3: Comparison of ASR WERs on the CoVoST 2 dataset between penalty parameter increase rates (IR) of 0.002 and 0.02 per epoch.

| Param | Lang | Two-stage (PT+FT) | VM-ASR UAS (IR=.02) | VM-ASR USA (IR=.02) | VM-ASR UAS (IR=.002) | VM-ASR USA (IR=.002) |
|---|---|---|---|---|---|---|
| | En | 29.7% | 26.8% | 27.1% | 29.3% | **25.7%** |
| | Fr | 26.5% | 24.3% | 24.7% | 26.1% | **22.6%** |
| 58M | De | 28.8% | 26.2% | 26.8% | 27.3% | **24.3%** |
| | Es | 21.3% | 18.9% | 19.1% | 20.2% | **17.5%** |
| | Ca | 18.2% | 16.2% | 16.5% | 17.9% | **14.3%** |
| | Ave. | 24.9% | 22.1% | 22.8% | 24.1% | **20.9%** |

This section examines the impact of MLO on ASR and S2TT performance through a comparison of Two-stage (PT+FT), static weight, joint PT+FT W/O MOO, VS-ASR, VC-ASR, and VM-ASR, with parameter sizes of 100M and 58M. For VM-ASR, we tested two optimization sequences: UAS (self-supervised $\rightarrow$ ASR $\rightarrow$ S2TT) and USA (self-supervised $\rightarrow$ S2TT $\rightarrow$ ASR).

**ASR Performance:** Table 1 highlights the superior performance of VM-ASR across languages and parameter sizes. For 100M-parameter models, VM-ASR (USA) achieves the lowest WER, outperforming Two-stage (PT+FT), Static Weight, Joint PT+FT without MOO, PEFT, VS-ASR, and VC-ASR by up to 22.3%. VM-ASR (UAS) shows comparable gains, improving by up to 23.7%. Similarly, for 58M-parameter models, VM-ASR (USA) achieves up to 11.9% improvement, while VM-ASR (UAS) achieves up to 14.6% improvement. These results affirm VM-ASR's effectiveness in multilingual, multitask ASR with its MLO strategy.

**S2TT Performance:** As shown in Table 2, VM-ASR also excels in S2TT tasks, achieving the highest BLEU scores for translations task. For 100M-parameter models, VM-ASR (USA) outperforms Two-stage (PT+FT), Static Weight, Joint PT+FT without MOO, PEFT, VS-ASR, and VC-ASR by up to 27.9%, with VM-ASR (UAS) achieving up to 25.1% improvement. For 58M-parameter models, VM-ASR (USA) achieves up to 20.5% improvement, while VM-ASR (UAS) achieves up to 18.9%. These consistent improvements demonstrate the robustness of VM-ASR in S2TT tasks.

## 5.2 CONFLICTING ASR AND S2TT OBJECTIVES

"Presence of multiple conflicting objectives degrades the model's performance."

In this section, we investigate the effect of conflicting objectives on the model's performance using two algorithm settings: PT+FT and VS-ASR, with different model sizes. In Appendix C, we investigate the presence of conflicting objectives in ASR in more detail.

Table 4: Comparison of S2TT average BLEU scores on the CoVoST 2 dataset between penalty parameter increase rates (IR) of 0.002 and 0.02 per epoch.

| Param | Lang→En | Two-stage (PT+FT) | VM-ASR UAS (IR=.02) | VM-ASR USA (IR=.02) | VM-ASR UAS (IR=.002) | VM-ASR USA (IR=.002) |
|---|---|---|---|---|---|---|
| 58M | Fr→En | 23.4 | 26.5 | 26.8 | **27.2** | 25.1 |
| | De→En | 15.0 | 17.1 | 17.4 | **17.6** | 16.2 |
| | Es→En | 24.0 | 25.6 | 25.9 | **25.8** | 25.3 |
| | Ca→En | 19.4 | 21.4 | 21.6 | **21.9** | 20.2 |
| | Ave. | 20.4 | 22.6 | 22.9 | **23.1** | 21.7 |

From Tables 1 and 2, the VS-ASR method consistently outperforms the PT+FT method. One major difference between these two methods is the use of MOO. Hence, this experiment indicates the presence of conflicts in the ASR and S2TT objectives and highlights the effectiveness of MOO in optimizing conflicting objectives.

## 5.3 OPTIMIZATION ORDER IN MULTILEVEL OPTIMIZATION

"The order of optimization significantly impacts ASR accuracy in Multilevel Optimization."

Here, we investigate the significance of optimization order in MLO for ASR. By comparing the performance of different ASR algorithms under varying optimization sequences (UAS and USA), we aim to elucidate how the optimization order affects ASR accuracy.

From the results in Table 1, we observe that the UAS optimization sequence consistently yields superior ASR performance compared to the USA (as the penalty parameter of the second level is gradually increased beyond 1), indicating the importance of prioritizing certain objectives in the training process. This finding underscores the optimization order when designing MLO methods for ASR. The same observation can be made from Table 2 for the S2TT task where the USA optimization sequence provides the best WER. This phenomenon is visually demonstrated in Figure 1.

## 5.4 EFFECT OF PENALTY PARAMETER

"Penalty parameters play a crucial role in MLO-based ASR training."

In penalty-based MLO problems, selecting the appropriate penalty parameter is crucial. These methods prioritize upper-level objectives while controlling lower-level objectives through a penalty term. Using a smaller penalty parameter can weaken constraint enforcement, causing suboptimal lower-level performance, slower convergence, and imbalanced optimization (Shen & Chen, 2023). This is evident in our multilingual multi-task ASR experiments. We further conducted experiments following the same training procedure as other simulations, using a 58M parameter model with two different penalty parameter increase rates. A lower increase rate of 0.002, capped at 1.5, resulted in worse WER for lower-level tasks, as shown in Tables 3 and 4. Given the equal importance of ASR

Table 5: ASR WERs (LibriSpeech) and CERs (AISHELL) for PT+FT, Joint PT+FT W/O MOO, VS-ASR, VC-ASR, and VM-ASR, with VM-ASR using UEC (self-supervised → English → Chinese) and UCE (self-supervised → Chinese → English) optimization sequences.

| Param | Lang | Two-stage (PT+FT) | Joint PT+FT W/O MOO | VS-ASR | VC-ASR | VM-ASR UEC | VM-ASR UCE |
|---|---|---|---|---|---|---|---|
| 100M | En | 6.2% | 5.9% | 6.1% | 5.7% | **5.2%** | 5.4% |
| | Zh | 6.0% | 5.6% | 5.8% | 5.5% | 5.3% | **5.0%** |
| | Ave | 6.1% | 5.7% | 5.9% | 5.6% | **5.2%** | 5.2% |
| 58M | En | 7.8% | 7.1% | 7.3% | 6.8% | **6.5%** | 6.6% |
| | Zh | 7.4% | 6.8% | 7.0% | 6.5% | 6.1% | **5.8%** |
| | Ave | 7.6% | 6.9% | 7.1% | 6.6% | 6.3% | **6.2%** |

and S2TT objectives in our study, we applied a larger penalty parameter increase rate of 0.02 for the lower levels, with a final value capped at 1.5. This adjustment improved lower-level performance but slightly degraded upper-level performance. Therefore, selecting the penalty parameter requires careful consideration of the trade-offs between upper- and lower-level priorities. A detailed explanation of the process of the selection of penalty parameter is provided in the Appendix: F.

## 5.5 OUR OBSERVATIONS PERSIST ACROSS DIFFERENT MODEL SIZES

"Our observations are consistent across different model sizes."

We assess the consistency of our observations across different model sizes (100M and 58M parameters) by evaluating ASR and S2TT performance. Results from Tables 1 and 2 confirm the reliability and generalizability of our findings, offering insights for scalable ASR system design.

**ASR Performance.** From Table 1 we observed that the PT+FT approach achieved competitive performance across all languages. The VS-ASR method consistently outperformed the PT+FT method. The VC-ASR model demonstrated even better performance. The most notable finding is the performance of the VM-ASR model, which exhibited significant improvements over other models. The VM-ASR model optimized with the UAS objective sequence achieved the lowest average WER, demonstrating its effectiveness in leveraging unlabeled data for improved performance. These observations are valid for both the 100M and the 58M parameter models.

**S2TT Performance.** Table 2 illustrates the S2TT WERs comparison for different models. Similar to ASR, the PT+FT approach demonstrated competitive performance across all language pairs. The VS-ASR model consistently outperformed other models in the S2TT task. Interestingly, the VM-ASR model optimized with the USA objective sequence achieved the lowest average WER, outperforming other models in the S2TT task. Both the 100M parameter model and the 58M parameter model demonstrate similar improvements.

## 5.6 CONSISTENT OBSERVATIONS IN LANGUAGE-BASED MLO

To verify our findings, we conduct experiments in both task-based and language-based MLO settings using similar hyperparameters for the LibriSpeech and AISHELL datasets. In this experiment, we use a combined dataset of LibriSpeech and AISHELL to perform MLO based on language types, focusing exclusively on the ASR task. The results, shown in Table 5, reveal a phenomenon similar to what we observed in task-based MLO. This consistent observation further validates our conclusion regarding the effectiveness of MLO in optimizing ASR tasks.

## 6 CONCLUSIONS AND LIMITATIONS

In conclusion, our study highlights the substantial advantages of integrating self-supervised loss as constraining objectives within a multilevel multi-objective optimization (MOO) structure for multilingual multi-task ASR training. Our findings strongly indicate that segregating highly conflicting objectives into different optimization levels yields significant benefits for ASR and S2TT tasks. This approach not only enhances the effectiveness of MOO but also underscores its potential for optimizing complex tasks across diverse linguistic boundaries. While our results are based on extensive simulations, further theoretical analysis would be an interesting direction for future research.

## 7 FUTURE WORK

Our study demonstrates the effectiveness of MOO methods in addressing conflicting objectives for multilingual, multi-task ASR and S2TT tasks. However, there remain areas for further exploration. Specifically, we hypothesize that when objectives are highly conflicting, their optimal solutions are far apart in the parameter space, resulting in a large and spread-out Pareto front that represents diverse trade-offs. Investigating this hypothesis and its implications for optimization strategies, particularly in highly conflicting scenarios, would provide deeper insights into managing such trade-offs effectively.

## 8 REPRODUCIBILITY STATEMENT

We document implementation details in Section 5 and Appendix E. The code is included in the supplemental materials, and we will publish it on GitHub upon acceptance of the paper.

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

# Supplementary Material for " Objective Soups: Multilingual Multi-Task Acoustic Modeling for Speech Processing "

# Table of Contents

## A   ALGORITHM DEVELOPMENT

After formalizing the three training algorithms in 3.2, our subsequent objective is to devise a gradient-based algorithm capable of addressing large-scale, high-dimensional multilingual multi-task challenges while ensuring guaranteed convergence to Pareto stationary solutions. We will focus on the algorithm development of VC-ASR, as this can be easily extended to the other two methods (VS-ASR, VM-ASR). To achieve a gradient-based algorithm for VC-ASR that can avoid conflicting update directions, we leverage recent advances in unconstrained MOO (Chen et al., 2023) and employ a penalty-based approach to convert the constrained MOO problem in 3 into an unconstrained MOO problem. This approach simultaneously conducts self-supervised pre-training and supervised multi-objective learning, as defined in Equation (3); that is,

$$\min_{\theta \in \mathbb{R}^s, \phi \in \mathbb{R}^r} L_\eta(\Theta) := [l_{\text{ctc}}(\theta, \phi_{1,1}) + \eta l_{\text{u}}(\theta), \cdots, l_{\text{ctc}}(\theta, \phi_{1,N}) + \eta l_{\text{u}}(\theta), \ldots, \quad (9)$$

$$l_{\text{ctc}}(\theta, \phi_{T,1}) + \eta l_{\text{u}}(\theta), \cdots, l_{\text{ctc}}(\theta, \phi_{T,N}) + \eta l_{\text{u}}(\theta)]$$

where $\eta$ is a penalty parameter. This penalty parameter integrates the self-supervised constrained objective with the supervised objectives and ensures that the feasible region of the supervised objective remains within certain bounds.

**Limitation of static weighting.** To guarantee Pareto stationary for supervised objectives, we can employ either static or dynamic weighting MOO methods. In static weighting, we optimize the (weighted) average of the multiple objectives (Kurin et al., 2022; Xin et al., 2022). This method is simple but it may suffer from conflicting objectives where gradients of multiple objectives may have conflicting directions. For instance, considering $l_{t,n}(\Theta) = l_{\text{ctc}}(\theta, \phi_{t,n}) + \eta l_{\text{u}}(\theta)$ and $l_{t',n'}(\Theta) = l_{\text{ctc}}(\theta, \phi_{t',n'}) + \eta l_{\text{u}}(\theta)$ two objectives having conflicting directions, $(t, t') \in [T]$ and $(n, n') \in [N]$, then $\langle \nabla_\Theta l_{t,n}(\Theta), \nabla_\Theta l_{t',n'}(\Theta) \rangle < 0$.

| Notation | Description |
|---|---|
| $\Theta \in \mathbb{R}^q$ | Model parameter including backbone and classification head parameter. |
| $\theta \in \mathbb{R}^s$ | Backbone parameter. |
| $\theta^k \in \mathbb{R}^s$ | Backbone parameter at $k$-th iteration. |
| $\theta^* \in \mathbb{R}^s$ | Optimum backbone parameter. |
| $\phi \in \mathbb{R}^r$ | Parameter of the task-specific classification head. |
| $\phi_{t,n} \in \mathbb{R}^r$ | Classification head parameter of $n$-th task and $t$-th language. |
| $\phi_{t,n}^k \in \mathbb{R}^r$ | Classification head parameter of $n$-th task and $t$-th language at $k$-th iteration. |
| $\phi_p \in \mathbb{R}^r$ | A group of all classification head parameters of level $p$. |
| $\phi^* \in \mathbb{R}^r$ | Optimum parameter of the task-specific classification head. |
| $L$ | Vector of all objectives. |
| $L_\eta$ | Vector of all objectives with penalized lower level constrained objective used for VC-ASR method. |
| $L_p$ | Vector of all objectives in level $p$ used for VM-ASR method. |
| $l_m, m \in [M]$ | $m$-th objective. |
| $l_{\mathrm{ctc}}$ | CTC loss with supervised data. |
| $l_{\mathrm{u}}$ | self-supervised loss. |
| $t \in [T]$ | Represents a specific language (For example: English, German, etc.). |
| $n \in [N]$ | Represents a specific task (For example: ASR or S2TT.). |
| $k \in [K]$ | Current iteration number. |
| $p \in [P]$ | Optimization level. |
| $\epsilon$ | Constraint defines the feasible region for the upper-level objectives |
| $d$ | Conflict avoiding update direction. |
| $\gamma$ | Learning rate of $\lambda$ update. |
| $\alpha$ | Learning rate of backbone parameter. |
| $\beta$ | Learning rate of task-specific classification parameter. |
| $\lambda$ | Dynamic weight to combine gradient. |
| $\lambda^k$ | Dynamic weight at $k$-th iteration. |
| $\lambda_u^k$ | Dynamic weight of self-supervised objective at $k$-th iteration. |
| $\lambda_{t,n}^k$ | Dynamic weight of $n$-th task and $t$-th language at $k$-th iteration. |
| $\lambda^*$ | Optimum dynamic weight to combine gradient. |
| $\eta_{p-1}, p \geq 2$ | Penalty parameter of $p$-th level of multilevel optimization (VM-ASR). |
| $\eta = \eta_p \times \eta_{p-1}$ | Combined penalty constant for the lowest level (VM-ASR). |
| $\zeta^k$ | Stochastic unlabeled sample during training at iteration $k$. |
| $\xi^k$ | Stochastic labeled sample during training at iteration $k$. |
| $D$ | Labeled dataset. |

Table 6: List of notations used in this paper

**Proposed dynamic weighting.** To avoid conflicting directions we can employ dynamic weighting method which uses dynamically weighted gradients from individual objectives to avoid conflict and enables optimization in conflict-avoiding (CA) direction (Chen et al., 2023). Specifically, a CA direction $d$ is the steepest common descent direction that maximizes the worst descent, given by

$$d(\Theta) = \arg\max_d \min_{\lambda \in \Delta^{NT}} -\langle \nabla L_\eta(\Theta)\lambda, d \rangle - \frac{1}{2}||d||^2. \tag{10}$$

By reformulation, such a direction is equal to dynamically weighted gradients of different objectives (Chen et al., 2023), given by $d(\Theta) = -\nabla L_\eta(\Theta)\lambda^*(\Theta)$ with weight $\lambda^*(\Theta)$ computed by

$$\lambda^*(\Theta) = \arg\min_{\lambda \in \Delta^{NT}} ||\nabla L_\eta(\Theta)\lambda||^2. \tag{11}$$

However, finding the true gradients of $\nabla L_\eta(\Theta)$ is computationally expensive. Hence, in our problem, we employ a stochastic variant of MGDA, the MoDo algorithm (Chen et al., 2023), which obtains an unbiased stochastic gradient estimate for (11) via a double sampling technique.

At each iteration $k$, denote $\xi_1^k$ and $\xi_2^k$ as two independent samples from labeled dataset $D$, and $\nabla l(\xi_1^k; \Theta^k)$ and $\nabla l(\xi_2^k; \Theta^k)$ as the stochastic gradients. We leverage the MoDo update in (Chen et al., 2023) by

$$\lambda^{k+1} = \Pi_{\Delta^{NT}}\left(\lambda^k - \gamma^k\left(\nabla L_\eta(\xi_1^k; \Theta^k)^\top \nabla L_\eta(\xi_2^k; \Theta^k)\right)\lambda^k\right) \tag{12}$$

where $\gamma^k$ is step size, $\Pi_{\Delta^{NT}}(\cdot)$ denotes the projection to $\Delta^{NT}$.

**Parameters update.** Using the dynamic weighting and penalization method, we update the backbone parameters, $\theta$, of the ASR model. Next, we describe the backbone parameters and task-specific classification parameters update rules for VS-ASR, VC-ASR, and VM-ASR.

**VS-ASR.** For single vectorized objective training, we only need to consider if the objectives have conflicting update directions. As in multilingual multi-task training we are using separate language dataset, we can assume that the objectives have conflicting update direction. We can also prove this assumption by calculating $\langle \nabla_\Theta l_{t,n}(\Theta), \nabla_\Theta l_{t',n'}(\Theta)\rangle < 0$. We optimize this vectorized objectives using algorithm: 1 where the shared backbone parameters are updated using the following equations,

$$\theta^{k+1} = \theta^k - \alpha \sum_{t=1}^{T}\sum_{n=1}^{N} \lambda_{t,n}^k \nabla_\theta l_{\mathrm{ctc}}(\theta^k, \phi_{t,n}^k) - \alpha\lambda_{\mathrm{u}}\nabla_\theta l_{\mathrm{u}}(\theta^k). \tag{13}$$

In this context, $\alpha > 0$ denotes the learning rate specifically assigned to the backbone parameters. Moreover, $\lambda_{t,n}^k$ and $\lambda_{\mathrm{u}}$ represent the dynamic update directions for supervised and self-supervised objectives, respectively, which are computed using the MoDo algorithm. Similarly, taking the gradients of each of the supervised objective functions with respect to parameters of task-specific output heads, task-specific output layers are updated via,

$$\phi_{t,n}^{k+1} = \phi_{t,n}^k - \beta \nabla_\phi l_{\mathrm{ctc}}(\phi_{t,n}^k, \theta^k) \tag{14}$$

where $\beta > 0$ is the learning rate for the task-specific parameter.

**VC-ASR.** To train a model using VC-ASR algorithm, the backbone parameters $\theta$ is updated using,

$$\theta^{k+1} = \theta^k - \alpha \sum_{t=1}^{T}\sum_{n=1}^{N} \lambda_{t,n}^k \nabla_\theta l_{\mathrm{ctc}}(\theta^k, \phi_{t,n}^k) - \alpha\eta\nabla_\theta l_{\mathrm{u}}(\theta^k). \tag{15}$$

To update task-specific classification heads, we employ (14); see a summary in Algorithm 2.

**VM-ASR.** In VM-ASR, we separate highly conflicting objectives into distinct optimization levels. Here, we assume that all objectives at level $p$ function as lower-level objectives for those at level $p-1$. Consequently, we can update the backbone parameters using the penalize method, that is

$$\theta^{k+1} = \theta^k - \alpha \sum_{t_1=1}^{T_1}\sum_{n_1=1}^{N_1} \lambda_{t_1,n_1}^k \nabla_\theta l_{\mathrm{ctc}}(\theta^k, \phi_{t_1,n_1}^k) - \alpha\eta_2\left(\sum_{t_2=1}^{T_2}\sum_{n_2=1}^{N_2} \lambda_{t_2,n_2}^k \nabla_\theta l_{\mathrm{ctc}}(\theta^k, \phi_{t_2,n_2}^k) + \cdots\right. \tag{16}$$

$$\alpha\eta_{p-1}\left(\sum_{t_p=1}^{T_p}\sum_{n_p=1}^{N_p} \lambda_{t_p,n_p}^k \nabla_\theta l_{\mathrm{ctc}}(\theta^k, \phi_{t_p,n_p}^k) + \cdots \alpha\eta_{P-1}\left(\sum_{t_P=1}^{T_P}\sum_{n_P=1}^{N_P} \lambda_{t_P,n_P}^k \nabla_\theta l_{\mathrm{ctc}}(\theta^k, \phi_{t_P,n_P}^k) + \alpha\eta\nabla_\theta l_{\mathrm{u}}(\theta^k)\right)\right).$$

Update task-specific classification parameters using

$$\phi_{t_p,n_p}^{k+1} = \phi_{t_p,n_p}^k - \beta \nabla_\phi l_{\mathrm{ctc}}(\phi_{t_p,n_p}^k, \theta^k) \tag{17}$$

where $N_p$ and $T_p$ represent the total number of tasks and languages at level $p$, respectively. We represent the penalty parameter at level $p$ as $\eta_p$ and for self-supervised objective, the penalty parameter is $\eta$.

## B  TASK SPECIFIC FORMULATION AND UPDATE RULE

In this section, we will explore in detail the three MOO setups in ASR and S2TT tasks and establish the parameter update rules for each of them.

## B.1 VS-ASR FOR SINGLE VECTORIZED OBJECTIVES

For single vectorized objective training, we only need to consider if the objectives have conflicting update directions. As in the multilingual multi-task training, we use separate language datasets, so we can assume that the objectives have conflicting update directions. We can also verify this assumption by calculating $\langle \nabla_\Theta l_{t,1}(\Theta), \nabla_\Theta l_{t',2}(\Theta) \rangle < 0$. We can formulate this single vectorized objectives for ASR and S2TT tasks following (2) as follows,

$$\min_{\Theta \in \mathbb{R}^q} [l_{ctc}(\theta, \phi_{1,1}), \cdots, l_{ctc}(\theta, \phi_{1,N}), \ldots, l_{ctc}(\theta, \phi_{T,1}), \cdots, l_{ctc}(\theta, \phi_{T,N}), l_u(\theta)]. \quad (18)$$

As there is no lower-level constrain, we optimize this vectorized objectives using algorithm: 1 where the shared backbone parameters are updated using the following equations

$$\theta^{k+1} = \theta^k - \alpha \sum_{t=1}^{T} \lambda_{t,1}^k \nabla_\theta l_{ctc}(\theta^k, \phi_{t,1}^k) - \alpha \sum_{t=1}^{T} \lambda_{t,2}^k \nabla_\theta l_{ctc}(\theta^k, \phi_{t,2}^k) - \alpha \lambda_u^k \nabla_\theta l_u(\theta^k) \quad (19)$$

where $\lambda_{t,1}$ and $\lambda_{t,2}$ are dynamic update directions for ASR and S2TT tasks, respectively, and $\lambda_u$ is the dynamic update direction for self-supervised objective calculated using MoDo algorithm. We update the classification heads using

$$\phi_{t,1}^{k+1} = \phi_{t,1}^k - \beta \nabla_\phi l_{ctc}(\phi_{t,1}^k, \theta^k). \quad (20a)$$

$$\phi_{t,2}^{k+1} = \phi_{t,2}^k - \beta \nabla_\phi l_{ctc}(\phi_{t,2}^k, \theta^k). \quad (20b)$$

## B.2 VC-ASR FOR VECTORIZED OBJECTIVES WITH CONSTRAINT LOWER LEVEL

In this setup, we use self-supervised CPC loss, $l_u(\theta)$, as a lower-level constraint to shrink the search region for the optimal Pareto stationary point for supervised CTC loss, $l_{ctc}(\theta, \phi)$. The problem formulation for VC-ASR in ASR and S2TT tasks can be written as follows:

$$\min_{\Theta \in \mathbb{R}^q} [l_{ctc}(\theta, \phi_{1,1}), l_{ctc}(\theta, \phi_{1,2}), \ldots, l_{ctc}(\theta, \phi_{T,1}), l_{ctc}(\theta, \phi_{T,2})]$$

$$\text{s.t. } l_u(\theta) - \min_\theta l_u(\theta) \leq \epsilon. \quad (21)$$

The backbone parameters $\theta$ is updated using,

$$\theta^{k+1} = \theta^k - \alpha \sum_{t=1}^{T} \lambda_{t,1}^k \nabla_\theta l_{ctc}(\theta^k, \phi_{t,1}^k) - \alpha \sum_{t=1}^{T} \lambda_{t,2}^k \nabla_\theta l_{ctc}(\theta^k, \phi_{t,2}^k) - \alpha \eta \nabla_\theta l_u(\theta^k). \quad (22)$$

The task specific classification parameters are updated using (20a) and (20b)

## B.3 VM-ASR FOR MULTILEVEL ASR OPTIMIZATION

In MLO problem, there is a hierarchy of objectives. We can reformulate the multilingual multi-task ASR optimization task into different MLO problems based on the tasks, languages, or language families to which they belong. We study these set-ups and solve these optimization problems using penalty-based gradient descent method.

**Multilevel optimization based on tasks.** We can extend the ASR optimization problem into three levels based on the tasks: ASR, S2TT, and self-supervised task. We always place the self-supervised objective at the lowest level and optimize it first, as the optimization of all other objectives directly depends on the optimization of the self-supervised objective.

$$\underset{\phi_{1,1}, \cdots, \phi_{T,1} \in \mathbb{R}^r, \phi_{1,2}^*, \ldots, \phi_{T,2}^*, \theta^*}{\arg\min} L_{ctc}(\phi_{1,1}, \phi_{2,1}, \ldots, \phi_{1,2}^*, \phi_{2,2}^*, \cdots, \theta^*)$$

$$\text{s.t. } \phi_{1,2}^*, \cdots, \phi_{T,2}^* = \underset{\phi_{1,2}, \cdots, \phi_{T,2} \in \mathbb{R}^r, \theta^*}{\arg\min} L_{ctc}(\phi_{1,1}, \phi_{2,1}, \ldots, \phi_{1,2}, \phi_{2,2}, \cdots, \theta^*)$$

$$\text{s.t. } \theta^* = \underset{\theta \in \mathbb{R}^s}{\arg\min} \; l_u(\theta). \quad (23)$$

We apply a penalty-based method to convert this multilevel multi-objective optimization problem into a single-level optimization problem and apply dynamic MOO to update the parameters in a conflict-avoiding direction.

$$\theta^{k+1} = \theta^k - \alpha \sum_{t=1}^{T} \lambda_{t,1}^k \nabla_\theta l_{\text{ctc}}(\theta^k, \phi_{t,1}^k) - \alpha\eta_1 \left( \sum_{t=1}^{T} \lambda_{t,2}^k \nabla_\theta l_{\text{ctc}}(\theta^k, \phi_{t,2}^k) + \alpha\eta_2 \nabla_\theta l_{\text{u}}(\theta^k) \right).$$
(24)

Here, $\eta_1$ and $\eta_2$ are penalty parameters. We can combine $\eta_1$ and $\eta_2$ and get $\eta = \eta_1 \times \eta_2$ for self-supervised loss.

$$\theta^{k+1} = \theta^k - \alpha \sum_{t=1}^{T} \lambda_{t,1}^k \nabla_\theta l_{\text{ctc}}(\theta^k, \phi_{t,1}^k) - \alpha\eta_1 \sum_{t=1}^{T} \lambda_{t,2}^k \nabla_\theta l_{\text{ctc}}(\theta^k, \phi_{t,2}^k) - \alpha\eta \nabla_\theta l_{\text{u}}(\theta^k).$$
(25)

Next, we update the classification heads via

$$\phi_{t,1}^{k+1} = \phi_{t,1}^k - \beta \nabla_\phi l_{\text{ctc}}(\phi_{t,1}^k, \theta^k).$$
(26a)

$$\phi_{t,2}^{k+1} = \phi_{t,2}^k - \beta \nabla_\phi l_{\text{ctc}}(\phi_{t,2}^k, \theta^k).$$
(26b)

We provide a detailed algorithm of multilevel ASR optimization in 3. We also do experiment altering the optimization order of ASR and S2TT tasks.

**Multilevel optimization based on language.** We can also extend ASR optimization problem into multiple level based on languages

$$\underset{\phi_{1,1},\phi_{1,2}\in\mathbb{R}^r,\phi_{2,1}^*,\phi_{2,2}^*,...,\theta^*}{\text{argmin}} L_{\text{ctc}}(\phi_{1,1},\phi_{1,2},\phi_{2,1}^*,\phi_{2,2}^*,\cdots,\theta^*)$$

$$\ddots$$

$$\text{s.t.} \quad \phi_{T,1}^*, \phi_{T,2}^* = \underset{\phi_{T,1},\phi_{T,2}\in\mathbb{R}^r,\theta^*}{\text{argmin}} L_{\text{ctc}}(\phi_{1,1},\phi_{1,2},\ldots,\phi_{T,1},\phi_{T,2},\theta^*)$$

$$\text{s.t.} \quad \theta^* = \underset{\theta\in\mathbb{R}^s}{\text{argmin}} \quad L_{\text{u}}(\theta).$$
(27)

In this setup, we optimize all the objectives of one language in one optimization level and optimize other languages' objectives in other optimization levels. For simplicity of implementation, we will consider two languages. We can update the model parameters using the following penalty-based update rules

$$\theta^{k+1} = \theta^k - \alpha \sum_{n=1}^{N} \lambda_{1,n}^k \nabla_\theta l_{\text{ctc}}(\theta^k, \phi_{1,n}^k) - \alpha\eta_1 \sum_{n=1}^{N} \lambda_{2,n}^k \nabla_\theta l_{\text{ctc}}(\theta^k, \phi_{2,n}^k) - \alpha\eta \nabla_\theta l_{\text{u}}(\theta^k).$$
(28)

In this equation, $\eta_1$ and $\eta_2$ are penalty parameters. We can combine $\eta_1$ and $\eta_2$ to obtain $\eta = \eta_1 \times \eta_2$, which is used for the self-supervised loss. The parameter $N = 2$ represents the total number of tasks (in this experiment, ASR and S2TT). The terms $\lambda_{1,n}^k$ and $\lambda_{2,n}^k$ represent the dynamic update directions for languages 1 and 2, respectively, during the $k$-th iteration for task $n$.

Next, we update the classification heads via

$$\phi_{t,1}^{k+1} = \phi_{t,1}^k - \beta \nabla_\phi l_{\text{ctc}}(\phi_{t,1}^k, \theta^k).$$
(29a)

$$\phi_{t,2}^{k+1} = \phi_{t,2}^k - \beta \nabla_\phi l_{\text{ctc}}(\phi_{t,2}^k, \theta^k).$$
(29b)

In both task-based and language-based MLO, we alter the order of objectives at the optimization level to examine the effects of their arrangement. By doing so, we can better understand how the sequence of objectives influences the optimization process and outcomes.

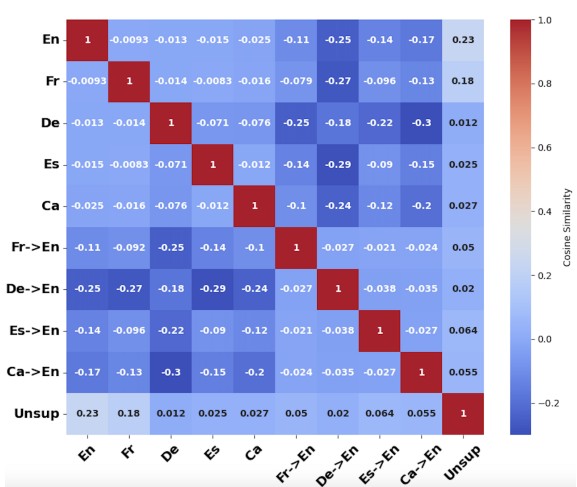

Figure 3: Heat-map of Cosine similarities among ASR and S2TT objectives.

---

**Algorithm 1** VS-ASR for multilingual multi-task ASR.

---

**Input:** Labeled data $(x, y)$, unlabeled data $X_{\mathrm{u}} := \{x_{\mathrm{u}}^1, x_{\mathrm{u}}^2, \cdots, x_{\mathrm{u}}^E\}$, learning rates $\alpha$ and $\beta$;
**for** $k = 1$ **to** $K$ **do**
    sample $\zeta_1^k = x_{1,\mathrm{u}}^k$, $\zeta_2^k = x_{2,\mathrm{u}}^k$, $\xi_1^k = (x_1^k, y_1^k)$ and $\xi_2^k = (x_2^k, y_2^k)$
    compute $\nabla l_{\mathrm{u}}(\zeta_1^k; \theta^k)$, $\nabla l_{\mathrm{u}}(\zeta_2^k; \theta^k)$, $\nabla l_{\mathrm{ctc}}(\xi_1^k; \theta^k, \phi^k)$, $\nabla l_{\mathrm{ctc}}(\xi_2^k; \theta^k, \phi^k)$
    update $\lambda^{k+1}$ by (12)
    update $\theta^{k+1}$ by (13)
    update $\phi_{t,n}^{k+1}$ by (14) $\forall t \in [T], \forall n \in [N]$
**end for**
**Output:** $\theta^K, \{\phi_{t,n}^K\}$

---

## C  GRADIENT CONFLICT

In this setup, we aim to separate highly conflicting objectives into upper and lower optimization levels. However, a sub-question arises within this setup: which objectives are highly conflicting? To address this question, we need to establish a boundary or threshold that distinguishes objectives with significant conflicts. We can create such a threshold by calculating the degree of conflict using the cosine similarity of the gradients of the objectives. If the cosine similarity of two objectives is smaller than a certain threshold, they are optimized at different levels. If $\nabla_\Theta l_{t,n}(\Theta)$ and $\nabla_\Theta l_{t',n'}(\Theta)$ are gradients of two objectives then we can calculate the cosine similarity using

$$\cos \omega = \frac{\langle \nabla_\Theta l_{t,n}(\Theta), \nabla_\Theta l_{t',n'}(\Theta) \rangle}{\|\nabla_\Theta l_{t,n}(\Theta)\| \|\nabla_\Theta l_{t',n'}(\Theta)\|} \tag{30}$$

where $\omega$ is the angle between the gradients of two different objectives. To calculate the similarity between update directions, we use the same conformer model and train it using two different languages and objectives simultaneously. We train the model for 20 epochs using both objectives and then average the gradients of their updates separately. We follow the same process for all languages and record their average gradients for 20 epochs. We can now calculate the cosine similarity between the gradient update direction of two objectives from these recorded gradients. We also compare the cosine similarity between self-supervised and supervised losses.

In Figure 3 and 4, we depict the cosine similarity of supervised objective gradients across five languages, along with the self-supervised objective gradient for ASR and S2TT. The heat map displays the similarity values, while the scatter plot, with points colored by their cluster assignments, helps visualize which objectives are closely related (high similarity) and which are not. The size and color of the points represent the similarity values and cluster assignments, respectively.

From the analysis of these figures, it is evident that tasks with lower similarities exhibit higher conflicts. Notably, the self-supervised gradients show significantly higher similarity with other objectives. This finding supports our decision to use the self-supervised loss as a lower-level constraint, thereby shrinking the search region for finding optimal Pareto points.

Moreover, segregating the highly conflicting ASR and S2TT tasks into different optimization levels reduced the overall conflict among the gradients of the objectives. Consequently, this approach improved the WER scores.

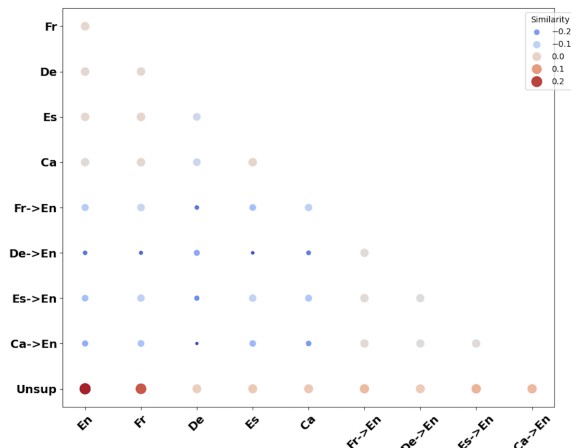

Figure 4: Scatter plot of cosine similarities between ASR and S2TT objectives.

---

**Algorithm 2** VC-ASR for multilingual multi-task ASR

---

> **Input:** Labeled data $(x, y)$, unlabeled data $X_{\mathrm{u}} \coloneqq \{x_{\mathrm{u}}^1, x_{\mathrm{u}}^2, \cdots, x_{\mathrm{u}}^E\}$, learning rates $\alpha, \beta$, and penalty parameter $\eta$;
> **for** $k = 1$ **to** $K$ **do**
>   sample $\zeta^k = x_{\mathrm{u}}^k$, $\xi_1^k = (x_1^k, y_1^k)$ and $\xi_2^k = (x_2^k, y_2^k)$
>   compute $\nabla l_{\mathrm{u}}(\zeta^k; \theta^k)$
>   compute $\nabla l_{\mathrm{ctc}}(\xi_1^k; \theta^k, \phi^k)$, $\nabla l_{\mathrm{ctc}}(\xi_2^k; \theta^k, \phi^k)$
>   update $\lambda^{k+1}$ by (12)
>   update $\theta^{k+1}$ by (15)
>   update $\phi_{t,n}^{k+1}$ by (14) $\forall t \in [T], \forall n \in [N]$
> **end for**
> **Output:** $\theta^K, \{\phi_{t,n}^K\}$

---

**Algorithm 3** VM-ASR for multilingual multi-task ASR.

---

> **Input:** Labeled data $(x, y)$, unlabeled data $X_{\mathrm{u}} \coloneqq \{x_{\mathrm{u}}^1, x_{\mathrm{u}}^2, \cdots, x_{\mathrm{u}}^E\}$, learning rates $\alpha, \beta$, and penalty $\eta_1, \cdots, \eta_P$;
> **for** $k = 1$ **to** $K$ **do**
>   sample $\zeta^k = x_{\mathrm{u}}^k$, $\xi_1^k = (x_1^k, y_1^k)$ and $\xi_2^k = (x_2^k, y_2^k)$
>   compute $\nabla l_{\mathrm{u}}(\zeta^k; \theta^k)$, $\nabla l_{\mathrm{ctc}}(\xi_1^k; \theta^k, \phi^k)$, $\nabla l_{\mathrm{ctc}}(\xi_2^k; \theta^k, \phi^k)$
>   update $\lambda^{k+1}$ by (12)
>   update $\theta^{k+1}$ by (16)
>   update $\phi_{t_p,n_p}^{k+1}$ by (17)$\forall t_p \in [T_p], \forall n_p \in [N_p]$
> **end for**
> **Output:** $\theta^K, \{\phi_{t,n}^K\}$

---

# D  BASELINE TRAINING METHODS

In this section, we outline the baseline methods used to compare against our MOO algorithms.

## D.1  PRE-TRAINING + FINE-TUNING (PT+FT)

This method involves two sequential steps:

**1. Pre-training:** The model is first pre-trained on a self-supervised learning (SSL) objective, such as CPC or Wav2Vec2, to learn general-purpose representations from unlabeled speech data. During this stage, the backbone parameters are updated using:

$$\theta^{k+1} = \theta^k - \alpha \nabla_\theta l_{\mathrm{u}}(\theta^k), \tag{31}$$

where $l_{\mathrm{u}}$ represents the SSL loss, and $\alpha$ is the learning rate.

**2. Fine-tuning:** After pre-training, the model is fine-tuned on a supervised task (e.g., ASR or S2TT) using the CTC loss to adapt the learned representations to task-specific objectives. During fine-tuning:

- The backbone parameters are updated using:

$$\theta^{k+1} = \theta^k - \frac{\beta}{NT} \sum_{t=1}^{T} \sum_{n=1}^{N} \nabla_\theta l_{\mathrm{ctc}}(\theta^k, \phi_{t,n}^k), \tag{32}$$

  where $\beta$ is the learning rate, $N$ and $T$ denote the number of tasks and languages, respectively.

- The parameters of the individual classification heads are updated using:

$$\phi_{t,n}^{k+1} = \phi_{t,n}^k - \beta \nabla_\phi l_{\mathrm{ctc}}(\phi_{t,n}^k, \theta^k), \tag{33}$$

  where $\phi_{t,n}$ denotes the parameters for task $n$ and language $t$.

### D.2 STATIC WEIGHTING

This method follows the same process as PT+FT but introduces static weighting during fine-tuning. Instead of using equal weights for all supervised objectives, a grid search is performed to assign suitable weights to each objective. The backbone parameters are updated using:

$$\theta^{k+1} = \theta^k - \beta \sum_{t=1}^{T} \sum_{n=1}^{N} \mu_{t,n} \nabla_\theta l_{\text{ctc}}(\theta^k, \phi_{t,n}^k), \tag{34}$$

where $\mu_{t,n}$ represents the static weight assigned to the supervised objective for task $n$ and language $t$. For our experiments, the following language-specific weights were used:

$$[\text{En, Fr, De, Es, Ca}] = [0.18, 0.19, 0.27, 0.16, 0.20].$$

#### JOINT PT+FT WITHOUT MOO

This method follows the same process as VC-ASR but does not incorporate MOO (Saif et al., 2024). Instead, all supervised objectives are optimized jointly without dynamic weighting or conflict-aware gradient alignment, resulting in a simpler optimization process.

### D.3 PARAMETER-EFFICIENT FINE-TUNING (PEFT)

In the PEFT method, the backbone is first pre-trained following Equation 31. Afterward, the backbone is frozen, and the fine-tuning is performed in a sequential manner:

1. A single set of classification heads is fine-tuned using Equation 33.
2. The fine-tuned classification heads are then frozen, and the next set is optimized.

This process continues iteratively for each set of classification heads.

## E EXPERIMENTAL SETUP

In this section, we outline the dataset, models, hyper-parameters, and data pre-processing techniques employed in evaluating our VS-ASR, VC-ASR, and VM-ASR algorithms.

**Dataset.** We evaluate our training algorithms on a combined dataset of LibriSpeech (Panayotov et al., 2015), AISHELL v1 (Bu et al., 2017), and CoVoST v2 (Wang et al., 2020). LibriSpeech is an English speech dataset consisting of 960 hours of data along with transcripts. AISHELL v1 is a 178-hour multi-channel Mandarin speech corpus designed for various speech/speaker processing tasks. We have combined these two datasets to create a single multilingual dataset. Our approach involved splitting the LibriSpeech dataset, allocating 860 hours for self-supervised pre-training and using the 100-hour train-clean-100 subset for supervised training. The trained models are tested on the AISHELL test dataset and the LibriSpeech test-clean dataset. During training using CoVoST dataset, we use equal batch sizes across all languages and tasks to ensure balanced training. For high-resource En, we fix a subset of data (top 50% from the provided CSV), while applying upsampling for low-resource languages—4x for Ca and Es and 2x for Fr and De. The same En subset is consistently used across all runs to maintain fairness.

In the first experiment, we use combined LibriSpeech and AISHELL multilingual dataset and train a multi-head conformer for multilingual ASR tasks. In the second experiment, we use the CoVoST v2 training dataset for multilingual ASR and S2TT training. The CoVoST v2 test set is used to evaluate the trained models. CoVoST v2 is a widely-used benchmark multilingual S2TT corpus covering translations from 21 languages into English and from English into 15 languages.

**Models.** We use two configurations of the Conformer model (Gulati et al., 2020), each with a different number of Conformer blocks and hidden units. The first model has 10 Conformer blocks with a hidden dimension of 612 units and 12 attention heads; the second model has 8 blocks with 512 hidden dimensions and 8 attention heads. Each attention head has a dimension of 51 for the first model and 64 for the second model. Both configurations use a convolutional kernel size of 31, enhancing the model's ability to discern temporal dependencies and capture long-range dependencies in the input

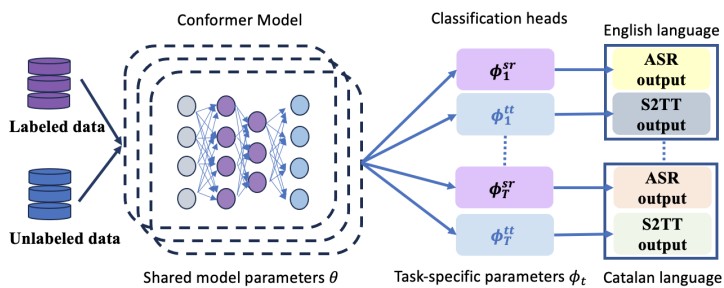

Figure 5: Multi-head conformer model for multilingual multi-task ASR.

sequences. Additionally, distinct classification heads are used for different tasks, each configured with varying output sizes (see Figure: 5).

**Hyper-parameters.** We use grid search to optimize hyperparameters, including learning rate, batch size, step size of MoDo, and penalty parameter increasing rate. For both SSL pre-training and supervised fine-tuning, the backbone learning rate is consistently set higher than the classification parameter learning rate. The SSL pre-training phase starts with a learning rate of $\alpha = 5 \times 10^{-4}$ for 100 epochs, annealed by a factor of 0.1 every 20 epochs. Fine-tuning uses a maximum learning rate of $\beta = 5 \times 10^{-5}$, with a scheduler reducing the learning rate by a factor of 0.1 if the test loss does not improve within 10 epochs. All MOO models (VS-ASR, VC-ASR, and VM-ASR) and joint PT+FT models are trained for 200 epochs. For PT+FT, we pre-train the model for 200 epochs and fine-tune it for an additional 100 epochs. A batch size of 256 and AdamW optimizer are used for both self-supervised and supervised training. The same hyperparameter settings are applied across all training methods to ensure consistency and comparability.

**Penalty parameter for ASR and S2TT.** For VC-ASR, the initial penalty parameter $\eta$ is set to 0 and increases at a rate of 0.02 per epoch. The increase stops once the penalty reaches a maximum value of 1.5. For VM-ASR, the second-level penalty parameter $\eta_1$ is initially set to 0.1 and increases by 0.02 per epoch, while the lower-level penalty constant $\eta_2$ starts at 0 and also increases by 0.02 per epoch. The increase for both penalty constants stops once they reach a maximum value of 1.5. A higher increase rate for the lower level ensures equal importance of both upper-level and lower-level objectives.

**Data pre-processing.** Our experiment involves both supervised and self-supervised training; however, preprocessing is applied only for the supervised training phase. For self-supervised training, we use raw speech data directly, enabling the model to learn representations from the audio without additional preprocessing. Specifically, we use a context length of 20 frames (200 ms) and predict the next 12 frames, employing 12 negative samples for contrastive loss. For supervised training, we apply standard preprocessing steps, including feature extraction and normalization. The raw audio files are converted into 80-dimensional log-mel features, a widely used representation in speech recognition tasks that effectively captures both temporal and spectral information. The data is then normalized to zero mean and unit variance to facilitate faster model convergence. We also employ SpecAug for data augmentation to improve model robustness. In terms of text processing, we utilize SentencePiece (Kudo & Richardson, 2018) as the tokenizer and detokenizer. We use word-based tokens, with the token vocabulary size set to 1000 for all languages except Chinese, where it is character-based with a vocabulary size of 5000. This ensures an appropriate balance between model complexity and performance. All training methods employ the same pre-processing steps.

**Computational Resources.** All simulations were run on two NVIDIA A5000 GPUs and two NVIDIA A4500 GPUs, with an Intel i9-7920X CPU and 128 GB of DDR4 memory.

# F ABLATION STUDY

In this section, we study the impact of different pre-training methods and provide a detailed explanation of the effect of the penalty parameter on the overall training process.

Table 7: ASR WERs and S2TT BLEU score comparison between CPC and BEST-RQ pre-training methods. For S2TT we do Lang → En translation.

| Param | Lang | VM-ASR-UAS (ASR-CPC-WER) | VM-ASR-UAS (S2TT-CPC-BLEU) | VM-ASR-UAS (ASR-BEST-RQ-WER) | VM-ASR-UAS (S2TT-BEST-RQ-BLEU) |
|---|---|---|---|---|---|
| | En | 23.5% | – | **21.8%** | – |
| | Fr | 16.0% | 30.9 | **14.9%** | **31.6** |
| 100M | De | 18.4% | 20.8 | **17.6%** | **21.4** |
| | Es | 14.1% | 30.1 | **13.2%** | **31.2** |
| | Ca | 11.6% | 25.8 | **10.8%** | **26.9** |
| | Ave. | 16.7% | 26.9 | **15.8%** | **27.8** |

Table 8: Comparison of ASR WERs and S2TT BLEU scores between Wav2Vec2 with and without VM-ASR methods. For S2TT, we perform translation from Lang → En.

| Param | Lang | Wav2Vec2-ASR Without VM-ASR | Wav2Vec2-S2TT Without VM-ASR | Wav2Vec2-ASR With VM-ASR | Wav2Vec2-S2TT With VM-ASR |
|---|---|---|---|---|---|
| | En | 19.4% | – | **17.9%** | – |
| | Fr | 14.1% | 32.4 | **12.8%** | **33.2** |
| 300M | De | 16.2% | 26.2 | **15.1%** | **27.7** |
| | Es | 11.1% | 33.7 | **9.7%** | **35.0** |
| | Ca | 9.8% | 28.1 | **8.9%** | **31.4** |
| | Ave. | 14.1% | 30.1 | **12.9%** | **31.8** |

## F.1 IMPACT OF PRE-TRAINING METHOD

In this ablation study, we assess the impact of two different pre-training techniques—CPC and BEST-RQ (Chiu et al., 2022)—on the performance of our VM-ASR method. The purpose of this ablation is to isolate the contribution of the pre-training method to the overall performance of the ASR and S2TT tasks. We keep the settings consistent across both methods, with the model containing 100 million parameters in all cases. The tasks evaluated include ASR in various languages and S2TT for translating from different source languages into English.

The results in Table 7 compare CPC and BEST-RQ across five languages. The results indicate a consistent improvement when using the BEST-RQ pre-training method. Specifically, BEST-RQ leads to a 5.4% absolute improvement in the average WER compared to CPC across all languages. The improvement is most pronounced in English and French, where the WER reductions reach 7.2% and 6.9%, respectively. For Spanish and German, the improvements are slightly smaller but still notable at 6.4% and 4.3%, respectively.

On the S2TT task, BEST-RQ also outperforms CPC, resulting in a 3.3% absolute increase in the average BLEU score across the evaluated languages. The highest BLEU score improvements are observed for Catalan and Spanish, with BEST-RQ providing increases of 4.3% and 3.7%, respectively. This indicates that BEST-RQ not only improves the ASR task but also enhances the downstream translation quality, likely due to the richer representations learned during pre-training.

Overall, these results suggest that the pre-training method plays a crucial role in enhancing both ASR and S2TT performance. The BEST-RQ approach, with its enhanced capability to model complex speech patterns, proves to be more effective than CPC, thus making it the more suitable choice for the VM-ASR algorithm.

## F.2 IMPACT OF VM-ASR ON FINE-TUNING SPEECH FOUNDATION MODEL

We evaluate our VM-ASR (UAS) method using the pre-trained Wav2Vec2-XLS-R[7] model (Babu et al., 2021). In this approach, we utilize the pre-trained model as the backbone and add linear layers for each task and language to predict the output vocabulary, training with the CTC method. All other

---

[7]https://huggingface.co/facebook/wav2vec2-xls-r-300m

Table 9: Comparison of resource requirements between a single MOO model and multiple single-objective models during deployment.

| Model | Encoder Param | Classification Heads | Total Param | Storage Size | Loading Time (s) |
|---|---|---|---|---|---|
| Single MOO Model | ~100M | ~2.5M | ~102.5M | ~654.4 MB | ~0.27 |
| Five Single-Objective Models | ~500M | ~2.5M | ~502.5M | ~3.2 GB | ~1.25 |

hyperparameters remain consistent with our previous training protocols. In the first experiment, we fine-tune the Wav2Vec2 model across all languages for ASR and S2TT tasks, following the same procedure as our earlier PT+FT training. In the second experiment, we perform joint pre-training and fine-tuning (PT+FT) with MOO, similar to the VM-ASR (USA) training approach. For both experiments, the model is trained for 50 epochs. The results are summarized in Table 8. On average, the Wav2Vec2 model trained with VM-ASR outperforms the standard Wav2Vec2 model by 8.5% in the ASR task and by 5.6% in the S2TT task.

### F.3 IMPACT OF PENALTY PARAMETER

In our multilingual multi-task ASR experiments, we investigated the effects of different penalty parameter increase rates to balance the ASR and S2TT tasks. We tested two configurations:

- A **lower increase rate of 0.002**, which led to worse WER/BLEU score for lower-level tasks, as shown in Tables 3 and 4.

- A **higher increase rate of 0.02**, which improved lower-level performance but slightly degraded upper-level performance.

**Choice of capped value for the penalty parameter:** We capped the penalty parameter at 1.5 based on our observed trade-off between upper- and lower-level tasks. A penalty higher than 1.5 could have improved lower-level performance further, but it would have significantly degraded upper-level metrics. Thus, 1.5 was chosen as an optimal balance point.

**Post-Maximum Penalty Effects:** The penalty parameter reached its maximum value of 1.5 after 75 epochs, but training continued for another 25 epochs. During this time, we observed further improvements in lower-level WER/BLEU scores, while upper-level performance deteriorated. This reinforces the critical role that penalty parameter selection plays in balancing competing objectives.

## G RESOURCE EFFICIENCY OF THE MOO MODEL

This section addresses the question: **How does a single MOO model reduce resource demands during deployment, making it a more efficient solution overall?**

- **Reduced Storage Requirements:** A single MOO model is highly memory-efficient due to parameter sharing across tasks, see Table: 9. The largest MOO model used in our experiments has a size of 654.4 MB, comprising an encoder (~100M parameters) shared across all objectives and five lightweight classification heads (~0.5M parameters each) for five different tasks (considering the ASR for five languages). In contrast, deploying five separate models for these tasks would require $5\times$ **more backbone parameters**, resulting in significantly higher storage demands. Assuming each single-objective model uses an encoder of similar size, the total storage requirement for separate models would reach approximately 3.2 GB.

- **Efficient Inference:** The MOO model also minimizes latency and computational overhead during inference. In our system, it takes only 0.27s to load the single MOO model, whereas loading five separate models takes 1.25s ($5 \times 0.25$ s). This reduction in loading time directly translates to faster response times and improved computational efficiency.

By consolidating multiple objectives into a single model, the MOO approach not only achieves significant memory savings but also ensures faster deployment and reduced computational demands, making it a scalable and efficient solution for real-world applications.

