# OpenReview forum: "Objective Soups: Multilingual Multi-Task Acoustic Modeling for Automatic Speech Recognition"
_ICLR.cc/2025/Conference — Submitted to ICLR 2025_

### Official Review · Reviewer_ZHdt · 2024-11-02

**Soundness:** 3
**Presentation:** 2
**Contribution:** 2
**Rating:** 5
**Confidence:** 4

**Summary:**

Multitask multilingual ASR consists of several objectives that may conflict with each other, and existing multiobjective optimization (MOO) methods can train models in a conflicting-objective-aware manner to resolve the conflicts to some extent. This paper provides a comprehensive comparative analysis of 3 MOO methods on 2 tasks (ASR and S2TT) and a suite of languages. The paper shows consistent improvements of MOO-based methods over non-MOO-based baselines and ablates several design choices.

**Strengths:**

1. The paper shows that training with an conflicting-objective-aware optimization technique (e.g. based on MOO) improves performance on ASR and S2TT across a bunch of languages, which was a nice finding.
2. The paper performs comprehensive analyses for each of their algorithm’s components e.g. the ordering of the tasks, the choice of the penalty hyperparameter, comparisons for 2 model sizes, etc.
3. The performance improvements are nice and consistent across settings.

**Weaknesses:**

1. The paper emphasizes multitask and multilingual learning but only considers 2 tasks (ASR and S2TT). It was unclear to me whether the findings of the paper would generalize to a true multitask setup with > 2 tasks. It does already consider several languages, which is great.
2. There are some missing baselines that I believe are needed to ensure that it is indeed conflicting objectives that cause degraded performance, and that MOO solves them. See the ‘Questions’ section for a description of these baselines.
3. Since the proposed methods are more resource-heavy than baseline, it would be fairer to report a memory-/speed-matched comparison between baselines and the proposed methods.
4. The related work section mentions that multi-task learning for joint ASR-S2TT has seen limited exploration but cites a range of papers that indeed do tackle joint ASR-S2TT; I recommend removing the ‘limited exploration’ phrase. Also, worth citing the Whisper paper (https://arxiv.org/abs/2212.04356) as an example of a model that does joint ASR and S2TT>
5. Writing: The writing can be made crisper as it was a little confusing at times. Concretely, throughout the paper you refer to ASR and S2TT tasks as ‘multitask ASR’; but S2TT is not an ASR task. I recommend calling them ‘multitask speech-to-text’ tasks for example. Also, the equations in Sections 4.1 - 4.3 consist of equations that are difficult to parse; I recommend simplifying them a little. The discussion in Section 5 can be made significantly crisper; there is no need to repeat many of the same numbers from the table again in the text.

**Questions:**

1. Each MOO objective (VS-ASR, VC-ASR, VM-ASR) seems to have a simpler non-MOO counterpart, which I refer to as the 'VS-ASR' baseline, 'VC-ASR' baseline and 'VM-ASR' baseline. I would like to see each of these baselines to ensure that the sophisticated MOO framework is actually what is resulting in the improved performance.
    1. 'VS-ASR' baseline: joint pretraining and finetuning (e.g. https://ieeexplore.ieee.org/abstract/document/10447834 which is cited in the paper). This baseline is already included in this paper, which is good.
    2. 'VC-ASR' baseline: pretraining followed by finetuning (by freezing the backbone, or by training the backbone with regularization on the pretraining loss), similar to KL divergence or L2 loss based continual learning regularizers.
    3. 'VM-ASR' baseline: pretraining, followed by freezing the backbone, finetuning one set of classification heads, freezing those and finetuning second set of classification heads, etc. (similar to parameter-efficient finetuning methods)
2. Could you describe the baselines (PT+FT, static weighting, and PT+FT w/o MOO) in more detail?

---

> ### Author Response · Authors · 2024-11-24
>
> We appreciate the reviewer’s recognition of our findings and analysis. Below, we address the concerns one by one.
>
> **W1. The paper emphasizes multitask and multilingual learning but only considers 2 tasks. It was unclear to me whether the findings of the paper would generalize to a true multitask setup. It does already consider several languages, which is great.**
>
> We appreciate the reviewer’s positive feedback on our work. Regarding the concern about generalization, we respectfully disagree. In the context of MOO, multitask learning can involve different tasks even if they come from different datasets [a]. In our experiment, we use five languages, each with two tasks (except English), which adds up to nine classification heads for nine distinct tasks. Therefore, we can confidently say that our methods handle a true multitask setup and can manage a substantial number of tasks.
>
> ### Reference
>
> [a] Chen et al., "Three-way trade-off in multi-objective learning: Optimization, generalization and conflict-avoidance," NeurIPS 2023.
>
> ----
> **W3. Since the proposed methods are more resource-heavy than baseline, it would be fairer to report a memory-/speed-matched comparison between baselines and the proposed methods.**
>
> That's an excellent idea! However, conducting a memory-/speed-matched comparison would require significant adjustments to both the baseline and our proposed MOO methods, which is beyond the current scope. We will consider this for future work to provide a more thorough evaluation.

---

> ### Author Response · Authors · 2024-11-24
>
> **W2 & Q1. Each MOO objective (VS-ASR, VC-ASR, VM-ASR) seems to have a simpler non-MOO counterpart, which I refer to as the 'VS-ASR' baseline, 'VC-ASR' baseline and 'VM-ASR' baseline, like to see each of these baselines.**
>
>  We appreciate your suggestion. The baselines for the VS-ASR and VC-ASR methods are already included in the paper as follows:
>
> **VS-ASR:** In this algorithm, we jointly trained all the conflicting objectives in a single optimization level using MOO. The **PT+FT method** or the **static weighting** method can be regarded as the baseline without MOO for VS-ASR (Please read the description of the PT+FT method presented in Q2).
>
> **VC-ASR:** In this algorithm, we used unsupervised objective as a constraint and used MOO to optimize all the conflicting objectives. **Joint PT+FT without MOO** is the baseline for the VC-ASR method where no MOO is used (Please read the description of the Joint PT+FT without MOO method presented in Q2).
>
> **VM-ASR:** Following your instructions, we trained the baseline for this method, which we term "parameter-efficient fine-tuning (PEFT)." In this PEFT method, the backbone is pre-trained, followed by freezing the backbone. Then, we fine-tune one set of classification heads, freeze those, and fine-tune the second set of classification heads, and so on.
>
>
> **Table-1:This table compares WERs of various ASR approaches, including PT+FT, Two-stage static, Joint PT+FT (W/O MOO), PEFT, VS-ASR, VC-ASR, and VM-ASR (with UAS and USA optimization sequences). Results are provided for two model sizes: 100M and 58M.**
>
>
> | Param | Lang | Two-stage (PT+FT) | Two-stage static | Joint PT+FT W/O MOO | PEFT | VS-ASR | VC-ASR | VM-ASR UAS | VM-ASR USA |
> |-------|------|--------------------|------------------|----------------------|-------|--------|--------|------------|------------|
> |   | En   | 26.8%             | 27.3%           | 25.2%               | 27.9% | 26.1%  | 24.6%  | **23.5%**  | 23.7%      |
> |       | Fr   | 19.6%             | 19.4%           | 17.8%               | 21.5% | 18.9%  | 17.1%  | **16.0%**  | 16.6%      |
> |    100M   | De   | 21.9%             | 21.8%           | 20.2%               | 23.8% | 21.2%  | 19.3%  | **18.4%**  | 18.5%      |
> |       | Es   | 17.8%             | 17.2%           | 15.9%               | 19.6% | 17.3%  | 15.2%  | **14.1%**  | 14.6%      |
> |       | Ca   | 14.3%             | 13.7%           | 13.1%               | 16.7% | 13.8%  | 12.5%  | **11.6%**  | 11.8%      |
> |       | **Ave.** | 20.1%      | 19.9%       | 18.4%           | 21.9% | 18.8% | 17.7% | **16.7%**  | 17.0%  |
> |    | En   | 29.7%             | 29.8%           | 28.4%               | 30.2% | 29.2%  | 27.9%  | **26.8%**  | 27.1%      |
> |       | Fr   | 26.5%             | 26.4%           | 25.9%               | 28.2% | 26.1%  | 25.2%  | **24.3%**  | 24.7%      |
> |    58M   | De   | 28.8%             | 28.6%           | 27.8%               | 30.1% | 28.3%  | 27.1%  | **26.2%**  | 26.8%      |
> |       | Es   | 21.3%             | 21.2%           | 20.4%               | 22.3% | 20.9%  | 19.4%  | **18.9%**  | 19.1%      |
> |       | Ca   | 18.2%             | 17.9%           | 17.5%               | 18.8% | 18.0%  | 16.9%  | **16.2%**  | 16.5%      |
> |       | **Ave.** | 24.9%      | 24.8%       | 24.0%         | 25.9% | 24.5% | 23.3% | **22.1%**  | 22.8%  |

---

> ### Author Response · Authors · 2024-11-24
>
> **Table-2: S2TT average(Ave.) BLEU score comparison using the CoVoST 2 dataset, including PT+FT, Two-stage static, Joint PT+FT (W/O MOO), PEFT, VS-ASR, VC-ASR, and VM-ASR (with UAS and USA optimization sequences). Results are provided for two model sizes: 100M and 58M.**
>
>
> | Param | Lang | Two-stage (PT+FT) | Two-stage static | Joint PT+FT W/O MOO | PEFT | VS-ASR | VC-ASR | VM-ASR UAS | VM-ASR USA |
> |-------|------|--------------------|------------------|----------------------|-------|--------|--------|------------|------------|
> |       | Fr → En   | 26.8             | 26.8           | 27.4               | 25.3 | 26.2  | 28.8  | 30.9  | **31.7**      |
> |   100M | De → En   | 17.4             | 17.5           | 18.9               | 15.9 | 18.1  | 19.9  | 20.8  | **21.5**      |
> |       | Es → En   | 26.1             | 26.3           | 27.3               | 24.7 | 27.0  | 28.2  | 30.1  | **30.6**      |
> |       | Ca → En   | 21.9             | 22.0           | 23.5               | 20.2 | 23.4  | 24.9  | 25.8  | **26.1**      |
> |       | **Ave.** | 23.0      | 23.1       | 24.3           | 21.5 | 23.7 | 25.4 | 26.9  | **27.5**  |
> |       | Fr → En   | 23.4             | 23.5           | 24.1               | 22.2 | 23.9  | 25.8  | 26.5  | **26.8**      |
> |   58M  | De → En   | 15.0            | 15.1           | 15.4               | 13.6 | 15.3  | 16.2  | 17.1  | **17.4**      |
> |       | Es → En   | 24.0            | 24.2           | 24.4               | 22.5 | 24.2  | 25.1  | 25.6  | **25.9**      |
> |       | Ca → En   | 19.4             | 19.2           | 19.2               | 17.9 | 19.1  | 20.3 | 21.4  | **21.6**     |
> |       | **Ave.** | 20.4      | 20.5       | 20.8           | 19.0 | 20.6 | 21.8 | 22.6  | **22.9**  |
>
>
> From Table-1 and 2, we can observe that PEFT's performance is worse than PT+FT and Joint PT+FT W/O MOO. The main reason is we kept the pre-trained backbone parameters frozen during the fine-tuning.
>
> **Q2. Could you describe the baselines (PT+FT, static weighting, and PT+FT w/o MOO) in more detail?**
>
> We appreciate your inquiry. Below, we provide a more detailed explanation of the baseline methods:
>
>
> **1. Pre-training + fine-tuning (PT+FT).** This baseline comprises two stages:
>
> **Pre-training.** The model undergoes pre-training on a self-supervised learning (SSL) objective, such as CPC or Wav2Vec2, to learn robust, general-purpose representations from unlabeled speech data. During this stage, the backbone parameters are updated using:
>
> $$
> \theta^{k+1} = \theta^{k} - \alpha\nabla_{\theta}l_{\rm u}(\theta^k),
> $$
>
> where $l_{\rm u}$ is the SSL loss, and $\alpha$ is the pre-training learning rate.
>
> **Fine-tuning.** Following pre-training, the model is fine-tuned on supervised tasks (e.g., ASR or S2TT) to adapt the learned representations. Fine-tuning utilizes the CTC loss across multiple tasks and languages.
>
> - **Backbone parameter update**:
>
> $$
> \theta^{k+1} = \theta^k - \frac{\beta}{NT} \sum_{t=1}^{T} \sum_{n=1}^{N} \nabla_\theta l_{ctc}(\theta^k, \phi_{t, n}^k)
> $$
>
> where $\beta$ is the learning rate, and $T$ and $N$ represent the number of tasks and languages, respectively.
>
> - **Classification head update**:
>
> $$
> \phi_{t, n}^{k+1} = \phi_{t, n}^k - \beta \nabla_\phi l_{\mathrm{ctc}}(\phi_{t, n}^k, \theta^k),
> $$
>
> where $\phi_{t, n}$ denotes the classification head parameters for task $t$ and language $n$.
>
> **2. Static weighting.** This method follows the same process as PT+FT but introduces static weighting during fine-tuning. Instead of equal weights for all supervised objectives, a grid search determines suitable weights for each objective. The backbone parameters are updated as:
>
> $$\theta^{k+1} = \theta^k - \beta \sum_{t=1}^{T} \sum_{n=1}^{N} \mu_{t, n} \nabla_\theta l_{ctc}(\theta^k, \phi_{t, n}^k),$$
>
> where $\mu_{t, n}$ is the static weight for task $t$ and language $n$.
>
> We used the following weights for different languages:
> $$[\text{En, Fr, De, Es, Ca}] = [0.18, 0.19, 0.27, 0.16, 0.20].$$
>
> **3. Joint PT+FT without MOO.**  This baseline is inspired by VC-ASR but excludes multi-objective optimization (MOO). All supervised objectives are optimized jointly without dynamic weighting or conflict-aware gradient alignment, resulting in a simpler optimization process.
>
> **4. Parameter-efficient fine-tuning (PEFT).** In the PEFT method, the backbone is pre-trained as described in PT+FT. During fine-tuning:
>
> 1. A single set of classification heads is fine-tuned using:
>    $$
>    \phi_{t, n}^{k+1} = \phi_{t, n}^k - \beta \nabla_\phi l_{\mathrm{ctc}}(\phi_{t, n}^k, \theta^k).
>    $$
>
> 2. The fine-tuned classification heads are frozen, and the process is repeated for the next set.
>
> This sequential optimization process ensures parameter efficiency by freezing previous sets while fine-tuning subsequent ones. We have added all the detailed descriptions in the Appendix-D.

---

> > ### Comment · Reviewer_ZHdt · 2024-11-25
> > **Response to Authors**
> >
> > Thank you for your response and the inclusion of the new PEFT baseline.

---

> > > ### Author Response · Authors · 2024-11-27
> > >
> > > Thank you very much for your prompt feedback. Please let us know if you have any further concerns or questions we can address.

---

### Official Review · Reviewer_TuPN · 2024-11-04

**Soundness:** 4
**Presentation:** 4
**Contribution:** 4
**Rating:** 10
**Confidence:** 4

**Summary:**

The paper shows advantages of integrating unsupervised loss as constraining objectives within a multilevel multi-objective optimization. The paper experimented with LibriSpeech and AISHELL and CoVoST v2 dataset for both ASR and speech-to-text translation tasks and has shown detailed results indicating segregating highly conflicting objectives into different optimization levels.

**Strengths:**

Novel approach in optimizing the multi-objective for ASR, S2TT tasks yielding improved results for Speech recognition, S2TT. The experiments and approach have been well-documented
It explained the reasoning of optimization order in MLO yielding different outcomes where UAS optimization sequence yields superior performance
It also shows that choosing right penalty parameter playing a crucial role in MLO based ASR training

**Weaknesses:**

The experimental results provided a stronger base for the results, however it could be helpful to have more theoretical reasoning for the same

**Questions:**

Can there be any further improvement done for improving training time or reducing GPU memory?

---

> ### Author Response · Authors · 2024-11-24
>
> We are glad that the reviewer finds our work novel and excellent. Below, we provide an answer to your question about computational complexity.
>
>
> **W1. The experimental results provided a stronger base for the results, however, it could be helpful to have more theoretical reasoning for the same.**
>
> Great suggestion! It is probably feasible to develop theoretical guarantees on the convergence rate of the studied algorithms. However, since we focus on empirical findings about how different approaches to handling multiple objectives help improve the performance of Multilingual Multi-Task ASR, it is beyond the scope of this paper. We thus leave it for future work.
>
> -------
>
> **Q1. Can there be any further improvement done for improving training time or reducing GPU memory?**
>
>
> Yes. The main computation bottleneck comes from the MOO algorithm that requires computing multiple objective gradients to find the conflict-avoidant (CA) direction. This bottleneck can be addressed by using the recently proposed approach Famo [a, Section 3.2]. Specifically, it uses an approximate method to compute the dynamic weight to compute the CA direction, without computing multiple objective gradients at each iteration, thus largely reducing both the training time and GPU memory.
>
>
> However, the improvement in computational complexity is beyond the scope of this paper, thus we leave it for future work.
>
> ### Reference
>
> [a] Liu et al., "Famo: Fast adaptive multitask optimization," NeurIPS, 2023.

---

### Official Review · Reviewer_KNnQ · 2024-11-04

**Soundness:** 3
**Presentation:** 3
**Contribution:** 3
**Rating:** 5
**Confidence:** 4

**Summary:**

This work explores the paradigm of multi-objective optimization (MOO) for multitask multilingual speech processing and proposes three multi-objective training objectives to improve ASR and speech-to-text translation (S2TT) across multiple languages. These objectives are designed by separating out the supervised/unsupervised/task-specific/language-specific losses into different optimization levels. The authors demonstrate that such multi-level optimization (MLO) is effective and provides significant performance gains compared to other baselines across multiple languages for both ASR and S2TT.

**Strengths:**

* The authors are the first to explore the use of multi-objective optimization at different optimization levels for ASR and S2TT across multiple languages.
* The paper is structured well and was easy to read.
* The authors derive consistent gains across both tasks and multiple languages using the proposed multi-level optimization technique.

**Weaknesses:**

* The reported baseline WERs and BLEU scores on CoVoST 2 appear to be significantly weaker than the state-of-the-art. The authors should clarify this further.

* The intuition for the proposed optimization levels could be motivated further. I elaborate on this point further in my questions to the authors below.

* The experimental section has many missing details that I elaborate in my questions below.

**Questions:**

* The baseline numbers for PT+FT shown in Tables 1 and 2 fall significantly short compared to prior work that uses Conformer models. For e.g., we can refer to the official repository accompanying the CoVoST 2 dataset (https://github.com/facebookresearch/fairseq/blob/main/examples/speech_to_text/docs/covost_example.md). WER on English using a 42.9M Conformer model (https://github.com/facebookresearch/fairseq/blob/main/examples/speech_to_text/docs/covost_example.md#results) is around 23%; the authors get 29.7% with their 58M parameter model. Similarly, on S2TT, the BLEU score for Fr->En using the 58M model in the official recipe is around 27 (https://github.com/facebookresearch/fairseq/blob/main/examples/speech_to_text/docs/covost_example.md#results-1), while the authors get a BLEU score of 23.4 with their 58M model. Please clarify this discrepancy in numbers.

* When motivating VC-ASR, the authors mention desirable properties of c(x) including the non-conflicting nature of its gradient update and that its optimization region must be sufficiently flat. We are referred to Appendix C for why the unsupervised loss possesses these properties. However, nothing is mentioned about the curvature of the loss (i.e., its flatness) in Appendix C. Please elaborate.


* Many experimental details are missing and should be added to an Appendix to encourage reproducibility. Below, a few missing details are listed:
  * In the sequential optimization of PT and FT objectives in the Joint PT+FT baseline, what learning rates were used?

  * What acoustic features were used, and what was the token vocabulary size?

  * Was any data augmentation (SpecAug) used during training?


* In Section 5.6, the authors should point to Appendix D that provides more details about the Librispeech/AISHELL experiment:

  * Why are numbers on test-other not reported in Table 5?

  * As is standard, are the numbers for AISHELL CERs and not WERs?  If so, this should be fixed in the caption of Table 5.

  * Table 5 is missing numbers with the Joint PT+FT baseline. How did this model fare?


* From Tables 1 and 2, UAS (VM-ASR) does better for ASR and USA (VM-ASR) does better for S2TT. Can the authors hypothesize why this might be? This trend of the task right after the unsupervised objective benefiting more also appears in Table 5 (UEC benefits En ASR more and UCE benefits Zh ASR more).


* Saif et al. 2024 is most related to this work and should be explained in more detail either in the "Related Work" section or in Section 3.2. The authors should explicitly mention how it differs from the proposed VS-ASR/VC-ASR objectives.


* Minor typo on line 185: It should be l_s(\theta, \phi_t^n) (not \phi_t^N).

---

> ### Author Response · Authors · 2024-11-24
>
> We appreciate the reviewer recognizing the strengths of our paper, particularly the benefits of MOO for multilingual speech-to-text tasks. Below, we address concerns one by one regarding baseline scores, the intuition behind optimization levels, and the experimental details.
>
> **W1 & Q1. Why do the baseline PT+FT numbers in Tables 1 and 2 fall short compared to prior Conformer-based work, such as the CoVoST 2 repository?**
>
> We respectfully acknowledge this difference and attribute the lower performance metrics to the following reasons:
>
> **Model architecture and setup:**  Unlike the GitHub Conformer models referenced, which use **separate models for each language and task**, we employ a **single unified model** for all languages and tasks. This approach increases training complexity because the model must balance diverse linguistic and task-specific objectives, often resulting in gradient conflicts during training.
>
> To illustrate, we compare the multi-objective training results with those of single-objective models, both of which have 58M parameters:
>
> | Model Type                 | Language | ASR Performance (WER) | S2TT Performance (BLEU) |
> |----------------------------|----------|------------------------|--------------------------|
> | **Single-objective (PT+FT)**| Catalan  | 13.6                    | 21.3                      |
> | **Multi-objective (PT+FT)**| Catalan  | 18.2                    | 19.4                      |
> | **Single-objective (PT+FT)**| Spanish  | 18.8                    | 26.1                      |
> | **Multi-objective (PT+FT)**| Spanish  | 21.3                    | 24.0                      |
>
>
> This comparison underscores the trade-offs between single-objective and multi-objective training setups, with the unified model facing challenges due to competing objectives but offering benefits in scalability and efficiency.
>
> --------
>
> **W2 & Q2. Nothing is mentioned about the curvature of the loss (i.e., its flatness) in Appendix C. Please elaborate.**
>
> Below, we clarify why the self-supervised loss $c(x)$ possesses the desirable properties, including flatness, and how it supports our motivation for VC-ASR.
>
> **Generalization and flatness**:
>    - As shown by [a], self-supervised learning methods optimize in **flatter loss regions**, indicated by smaller Hessian eigenvalues and lower sharpness metrics. Flat minima improve generalization and ensure stable optimization.
>    - For VC-ASR, this flatness allows $c(x)$ to **generalize features across tasks**, **reducing the search space for Pareto solutions** and aligning with conflicting objectives.
>
>    - Self-supervised losses provide robust feature representations that transfer well to supervised tasks, creating a stable foundation for multi-objective optimization.
>
> **Empirical Evidence**:
>    - To illustrate, we analyzed the sharpness of the loss landscape for supervised, PT+FT, and VM-ASR training. Curvature is measured using the largest eigenvalue of the Hessian $( |\lambda_{\text{max}}|)$ and sharpness-aware metrics like $(C_\epsilon, A)$-sharpness. As shown in the table below, the VM-ASR model consistently demonstrates lower curvature and better flatness compared to the supervised and PT+FT models.  And PT+FT method demonstrates lower curvature and better flatness compared to the supervised model.
>
> | Metric                  | Supervised model    | PT+FT     | VC-ASR |
> |-------------------------|------------|------------|------------|
> | $\|\lambda_{\text{max}}\|$      | 173.6 | 34.3 | 21.4    |
> | $(C_\epsilon, A)$-sharpness   | 78.5  | 2.9  | 1.2    |
>
>
> **Revisions to Appendix C**:
>    - We acknowledge the lack of explicit discussion on flatness in Appendix C. In the revision, we will incorporate insights from [a] on how self-supervised losses lead to flatter optimization regions and enhance gradient compatibility.
>
> ### Reference:
> [a] Fradkin et al., "Robustness to adversarial gradients: A glimpse into the loss landscape of contrastive pre-training," ICML 2022.

---

> ### Author Response · Authors · 2024-11-24
>
> **W3 & Q3. Many experimental details are missing and should be added to an Appendix to encourage reproducibility.**
>
> Thank you for pointing this out. We have included the following experimental details in the revised version of the paper:
>
> 1. **Learning Rates for PT and FT in Joint PT+FT Baseline:**
> - **Pre-training Learning Rate**: The unsupervised pre-training starts with a learning rate of $\alpha=5e-4$ for 100 epochs, which is then annealed by a factor of 0.1 every 20 epochs. The model is pre-trained for 200 epochs.
> - **Fine-tuning Learning Rate**: The maximum learning rate for fine-tuning is $\beta=5e -5$. A learning rate scheduler is used to monitor the test loss every 10 epochs. If the loss does not decrease during these 10 epochs, the learning rate is reduced by a factor of 0.1. The model is fine-tuned for a total of 100 epochs.
>
>
> 2. **Optimizer:**
> - **AdamW** for both pre-training and fine-tuning
>
> 3. **Acoustic Features and Token Vocabulary Size:**
> - **Pre-training**: We used **raw speech data** during pre-training to allow the model to learn representations directly from the audio. We utilize a context length of 20 frames (200 ms) and predict the next 12 frames, employing 12 negative samples for contrastive loss.
> - **Fine-tuning**: The raw audio files were converted into **80-dimensional log-mel features**. These features are commonly used in speech recognition tasks as they capture both temporal and spectral information effectively.
> - **Token Vocabulary Size**: The token vocabulary size was set to 1000 for all languages except Chinese, where it was set to 5000.
>
> 4. **Data Augmentation (SpecAugment):**
> - **Data Augmentation**: We employed **SpecAugment**, which includes **time** and **frequency masking**, during fine-tuning.
> All this training information has been added to the main paper's Appendix.
>
> ---
>
> **Q4. The authors should point to Appendix D that provides more details about the Librispeech/AISHELL experiment.**
>
> Thank you for pointing this out. We have added the information in Appendix D for more details on the LibriSpeech/AISHELL experiment.
>
> **Referencing Appendix D for More Experimental Details:**
> - The LibriSpeech/Aishell training follows the same procedures and hyperparameters as the CoVoST dataset training. We will clarify this setup in Appendix D.
>
> -------
>
> **Exclusion of Test-Other Results in Table 5:**
> - The **test-other results** were excluded from **Table 5** to allow for a more focused comparison with the AISHELL dataset. This exclusion helps us better understand the effect of multilevel optimization and MOO across different languages.
>
> Below is the results on **"test-other."**
>
> | Param   | Lang | Two-stage (PT+FT) | Joint PT+FT W/O MOO | VS-ASR | VC-ASR | VM-ASR (UEC) | VM-ASR (UCE) |
> |---------|------|-------------------|---------------------|--------|--------|--------------|--------------|
> | 100M    | En   | 17.0%              | 16.8%                | 17.1%   | 16.7%   | **16.3%**     | 16.5%         |
> | 58M     | En   | 17.8%              | 17.5%                | 17.7%   | 17.3%   | **17.0%**     | 17.1%         |
>
> ------
>
> **CER for AISHELL Dataset:**
>
> Thank you for your comment. Yes, we used CER to evaluate the AISHELL dataset and have already updated the caption of Table 5 to clarify this.
>
> -------
>
> **Joint PT+FT W/O MOO Baseline in Table 5:**
>
> Thank you for pointing that out. We have included the performance numbers for the Joint PT+FT baseline in the updated version of the paper. You can find the relevant results in the revised table.
>
>
> **Table-5: This table compares WERs for ASR tasks on the LibriSpeech and AISHELL datasets across various approaches, including Two-stage PT+FT, Joint PT+FT W/O MOO, VS-ASR, VC-ASR, and VM-ASR (using UEC and UCE sequences). Results are provided for two model sizes: 100M and 58M.**
>
>
> | Param | Lang | Two-stage (PT+FT) | Joint PT+FT W/O MOO | VS-ASR | VC-ASR | VM-ASR UEC | VM-ASR UCE |
> |-------|------|--------------------|---------------------|--------|--------|------------|------------|
> | 100M  | En   | 6.2%              | 5.9%               | 6.1%   | 5.7%   | **5.2%**   | 5.4%       |
> |       | Zh   | 6.0%              | 5.6%               | 5.8%   | 5.5%   | 5.3%       | **5.0%**   |
> |       | **Ave** | **6.1%**        | **5.7%**           | **5.9%** | **5.6%** | **5.2%** | **5.2%**   |
> | 58M   | En   | 7.8%              | 7.1%               | 7.3%   | 6.8%   | **6.5%**   | 6.6%       |
> |       | Zh   | 7.4%              | 6.8%               | 7.0%   | 6.5%   | 6.1%       | **5.8%**   |
> |       | **Ave** | **7.6%**        | **6.9%**           | **7.1%** | **6.6%** | **6.3%** | **6.2%**   |
>
> From the table, we can observe that Joint PT+FT W/O MOO performs better than PT+FT and VS-ASR methods.

---

> > ### Author Response · Authors · 2024-11-24
> >
> > **Q5. Why does UAS (VM-ASR) perform better for ASR and USA (VM-ASR) for S2TT, as seen in Tables 1 and 2? Also, why does the unsupervised objective seem to benefit subsequent tasks, as shown in Table 5?**
> >
> > We appreciate the comment and have the following hypothesis regarding the observed trend. The trend where the objective immediately above the unsupervised objective benefits more is likely due to the **larger penalty parameter increase rate** applied at that level. This rate prioritizes optimization for the objective directly following the unsupervised one [a]. As highlighted in Appendix-F3, the penalty parameter directly influences the trade-off between upper and lower levels, which aligns with the observed trends in both UAS (VM-ASR) for ASR and USA (VM-ASR) for S2TT.
> >
> > ### Reference
> > [a] Shen et al., "On penalty-based bilevel gradient descent method," ICML 2023.
> >
> > ------
> >
> > **Q6. Saif et al. 2024 is most related to this work and should be explained in more detail.**
> >
> > Indeed, [Saif et al. 2024] is related to this work. While there are similarities, our work is distinct in several important ways, as outlined below:
> >
> > - **VS-ASR**: In VS-ASR, we optimize all conflicting objectives within a single optimization level, meaning there are no explicit constraints in the optimization process. To address the conflicts, we employ MOO. This approach is fundamentally different from that of [Saif et al. 2024], as the method therein does not adopt this single-level optimization strategy.
> >
> > - **VC-ASR**: While there are some similarities between VC-ASR and [Saif et al. 2024], particularly in the presence of a lower-level constraint, our methods differ significantly in purpose and implementation:
> >   1. **Objective of the constraint**:
> >      - We use an unsupervised objective as a constraint to narrow the search space for identifying the most suitable Pareto optimal point on the Pareto front.
> >      - In contrast, [Saif et al. 2024] employ this constraint to establish a feedback loop between pre-training and fine-tuning, enabling the models to be jointly pre-trained and fine-tuned.
> >   2. **Upper-level optimization**:
> >      - In VC-ASR, we implement MOO in the upper-level to handle conflicts between objectives.
> >      - [Saif et al. 2024] do not experiment with MOO.
> >
> > In the revised version of the paper, we have explicitly highlighted these differences in section 3.2 to provide a clearer comparison.
> >
> > ---
> > **Q7. Minor typo on line 185: It should be $l_s(\theta, \phi_t^n)$ not $(\phi_t^N)$.**
> >
> > Thanks for pointing this out. We have corrected the typo in the main paper.

---

### Official Review · Reviewer_ECDc · 2024-11-04

**Soundness:** 3
**Presentation:** 2
**Contribution:** 3
**Rating:** 6
**Confidence:** 4

**Summary:**

This paper explores whether separating highly conflicting objectives into different optimization levels or keeping them unified is more effective for multilingual, multi-task learning in automatic speech recognition (ASR) and speech-to-text translation (S2TT). It introduces three Multi-Objective Optimization (MOO) methods—VS-ASR, VC-ASR, and VM-ASR—designed to resolve conflicts between objectives at various optimization stages. These approaches are compared against baseline methods like pre-training and fine-tuning (PT+FT) without MOO. Through experiments on LibriSpeech, AISHELL v1, and CoVoST v2 datasets, the study demonstrates that applying MOO across multiple optimization levels yields consistent improvements in both self-supervised and supervised tasks. A key insight is that segregating highly conflicting objectives enhances performance for ASR and S2TT, and the order of optimization significantly affects ASR accuracy. The study also emphasizes the importance of penalty parameters in Multi-Level Optimization (MLO) training, showing robustness across different model sizes. These findings offer valuable insights for optimizing conflicting objectives in multilingual and multi-task speech systems, with broader implications for the development of more efficient ASR and S2TT models.

**Strengths:**

1. Effective Handling of Conflicting Objectives: This paper directly addresses the issue of conflicting objectives by utilizing conflict-avoidant update directions, overcoming the limitations of traditional methods like static weighting and constrained optimization.
2. Comprehensive Exploration of MOO Techniques: Unlike related work (e.g., "reward soups" by Rame et al., 2024), which focuses on linear scalarization for Pareto-optimality, this paper explores a broader range of Multi-Objective Optimization (MOO) techniques to effectively manage conflicts in multilingual, multi-task learning.
3. Transparency and Reproducibility: The authors provide accessible code, ensuring reproducibility, and clearly outline their experimental setup, facilitating verification.
4. Thorough Evaluation: Experiments are conducted in both task-based and language-based MLO settings using consistent hyperparameters, and considerations such as memory consumption and training time are discussed, adding depth to the evaluation.

**Weaknesses:**

1. Code Quality: The provided code requires cleaning and refinement for better readability and maintainability.
2. Organization and Flow:
a. Important conclusions and interesting findings are relegated to the appendix, disrupting the reading flow.
b. Key sections (e.g., Section 5.2) should be integrated into the main text for coherence.
3. Abbreviation Clarity:
a. Certain abbreviations (e.g., FT, VC-ASR in the introduction) are used without proper introduction or explanation.
b. A clear glossary or introductory explanations would enhance readability.
4. Limited Evaluation:
a. Only two models (Conformer 58M and 100M) are evaluated.
b. Broader evaluation across diverse architectures and sizes is necessary to convincingly demonstrate the method's effectiveness.
5. Theoretical Analysis: Further theoretical analysis is needed (Notes as Future Work in Conclusion)  to comprehensively understand the capabilities and limitations of the proposed method.

**Questions:**

1. Clarification on Conflict Reduction: Could you provide more details on how segregating highly conflicting tasks (ASR and S2TT) into different optimization levels reduces overall conflict among gradients, leading to improved WER scores?
2. Efficiency of MOO Model: Section 5: Can you elaborate on how a single MOO model reduces resource demands during deployment, making it a more efficient solution overall? Are there specific metrics or comparisons that support this claim?
3. WER Score Discrepancy: It's surprising that en, fr, and de have higher WER despite having more data hours in CoVoST2. Can you offer insights into this discrepancy?
4. Pareto Front Explanation: Could you clarify the statement "In such scenarios, the Pareto front tends to be large and spread out"? How do these scenarios arise, and what implications does this have for the trade-offs between objectives?

---

> ### Author Response · Authors · 2024-11-24
>
> We appreciate the reviewer acknowledging our methods for handling conflicting objectives. Below, we address your questions and concerns one by one.
>
> **W1. The provided code requires cleaning and refinement for better readability and maintainability.**
>
> We appreciate your feedback and are working on improving the code to enhance its readability and maintainability. The updated version will be uploaded to GitHub soon.
>
> ----
> **W2. a. Important conclusions and interesting findings are relegated to the appendix.
> b. Key sections (e.g., Section 5.2) should be integrated into the main text.**
>
> We acknowledge that some interesting findings are included in the appendix due to page limitations. Our focus has been on presenting the most critical insights in the main paper, but we recognize the value of these additional results. We will try to incorporate some of these findings into the main paper in the revision.
>
> -----
>
> **W4. Limited model evaluation; broader testing across architectures and sizes is needed for stronger validation.**
>
> We focus on the Conformer model as it is a widely used ASR baseline, ensuring relevance to current research. To provide a broader evaluation, we have included Wav2Vec2 results in Appendix F, where our methods also outperform the baselines.
>
> ----
>
> **W5. Further theoretical analysis is needed (Notes as Future Work in Conclusion) to comprehensively understand the capabilities and limitations of the proposed method.**
>
> We appreciate your suggestion. Currently, our focus is primarily on improving the performance of the ASR and S2TT tasks. In the future, we plan to conduct a detailed theoretical analysis, including providing guarantees on the convergence rate of the studied algorithms, to gain a deeper understanding of the capabilities and limitations of our proposed methods.

---

> ### Author Response · Authors · 2024-11-24
>
> -----
>
> **Q1. How separating conflicting tasks into different optimization levels reduces gradient conflict and improves WER?**
>
>
>
> To mitigate gradient conflicts we separate ASR and S2TT (VM-ASR) tasks into distinct optimization levels, thereby reducing conflicts among their objectives. This hierarchical approach first optimizes objectives at the lower levels before addressing those at the upper levels, rather than simultaneously optimizing all objectives. By doing so, we minimize the likelihood and magnitude of conflicts arising from competing objective gradients.
>
> Moreover, increasing the number of optimization levels for a fixed total number of objectives reduces the average number of objectives per level. With fewer objectives at each level, the probability of gradient conflicts decreases, and the magnitude of maximum gradient conflicts between any pair of objectives at that level becomes smaller.
>
> To substantiate this claim, we provide quantification of gradient conflicts across different methods below.
>
> **Quantification of gradient conflicts:**
>
> - To quantify the conflict between tasks, we calculate the cosine similarity of gradients for different pairs of losses in VC-ASR and VM-ASR training:
>   - $$ \cos \omega_{A,S} = \frac{\langle \nabla L_{\text{ASR,CTC}}, \nabla L_{\text{S2TT,CTC}} \rangle}{\|\nabla L_{\text{ASR,CTC}}\| \|\nabla L_{\text{S2TT,CTC}}\|}$$
>   - $$\cos \omega_{A,C} = \frac{\langle \nabla L_{\text{ASR,CTC}}, \nabla L_{\text{CPC}} \rangle}{\|\nabla L_{\text{ASR,CTC}}\| \|\nabla L_{\text{CPC}}\|}$$
>   - $$\cos \omega_{S,C} = \frac{\langle \nabla L_{\text{S2TT,CTC}}, \nabla L_{\text{CPC}} \rangle}{\|\nabla L_{\text{S2TT,CTC}}\| \|\nabla L_{\text{CPC}}\|}$$
>
> - A **lower cosine similarity (negative)** indicates **greater conflict** between gradients, while **higher (positive)** values suggest **better alignment**.
> - By comparing these values in single-level (VS-ASR) and multilevel (VM-ASR) training, we observe that **multilevel training** consistently results in **higher similarity values**, demonstrating its ability to reduce conflicts effectively.
> - We also calculate the Number of Conflicts in one epoch, which represents the number of batches where significant gradient conflicts occur ($\langle \nabla L_{\text{t1}}, \nabla L_{\text{t2}} \rangle < 0$). Out of the total 1498 batches, 316 and 128 batches exhibit conflicts in VS-ASR and VM-ASR, respectively. The reduced conflicts in VM-ASR highlight its effectiveness in aligning gradients across tasks, resulting in more stable and efficient training.
>
>
> The cosine similarity of the gradients of the losses and the number of conflicts were calculated after 200 epochs on the CoVoST dataset, with consistent training hyperparameters across both setups.
>
> | Optimization type | $\cos \omega_{A,S}$ | $\cos \omega_{A,C}$ | $\cos \omega_{S,C}$ | Number of Conflict in one epoch | Stability |
> |-------------------|---------------------|---------------------|---------------------|-------------------------------------|-----------|
> | Single-level (VS-ASR) | 0.07 | 0.11 | 0.09 | 316 | Lower |
> | Multilevel (VM-ASR) | 0.12 | 0.17 | 0.14 | 128 | Higher |
>
> ### Reference
>
> [a] Sato et al., "A gradient method for multilevel optimization. Advances in Neural Information Processing Systems," NeurIPS 2021.
>
> [b] Saif et al., "Joint Unsupervised and Supervised Training for Automatic Speech Recognition via Bilevel Optimization," ICASSP 2024.
>
> **Q2. How a single MOO model reduces resource demands during deployment, making it a more efficient solution overall?**
>
> Thank you for your insightful question. A single MOO model can reduce resource demands during deployment by:
>
> 1. **Reduced storage requirements:** The MOO model (654.4 MB) handles five languages, with an encoder size of approximately 100M and each classification head about 0.5M. In the MOO setup, backbone parameters are shared across all objectives, making it a memory-efficient method. In contrast, deploying five separate models for five different tasks would require **$5\times$ more backbone parameters**, assuming each single-objective model uses the same encoder size as the MOO model.
> 2. **Efficient inference:** Loading the MOO model takes 0.27 seconds, whereas loading five separate models takes 1.25 seconds $(5 \times 0.25s)$, significantly reducing latency and overhead.
>
> | Model                          | Encoder Parameters | Classification Heads | Total Parameters | Storage Size | Loading Time |
> |--------------------------------|--------------------|-----------------------|------------------|--------------|--------------|
> | **Single MOO model**           | ~100M             | ~$5 \times 0.5$M = ~2.5M    | ~102.5M          | 654.4 MB     | 0.27 s       |
> | **Multiple single-objective models** | ~$5 \times 100$M = 500M | ~$5 \times 0.5$M = ~2.5M    | ~502.5M          | ~3.2 GB        | ~$5 \times 0.25$ s  |

---

> ### Author Response · Authors · 2024-11-24
>
> **Q3. Why En, Fr, and De have higher WER despite having more data hours in CoVoST2?**
>
> We believe that the higher WER for En, Fr, and De, despite having more data hours in CoVoST2, can be explained by:
>
> **Quality of test dataset:** Similar to our paper, the CoVoST 2 paper also reported high WER and low BLEU scores for En, Fr, and De. See [a]. These discrepancies may be attributed to the use of more challenging test datasets.
>
> ### Reference
>
> [a] Wang et al., "CoVoST 2 and Massively Multilingual Speech Translation," Interspeech 2021.
>
> -----
>
> **Q4. Could you clarify the statement "In such scenarios, the Pareto front tends to be large and spread out"? How do these scenarios arise, and what are the implications?**
>
> Thanks for the question. Indeed, this is only an intuition based on our empirical studies which we detail below.
>
> **Intuition:** when objectives are **highly conflicting**, the optimal vector-valued objectives can be **far apart** in the objective space. As a result, the Pareto front could become **wide and spread out**.
>
> We will clarify in the revised paper that **this is our intuition and needs further verification** and move this sentence to future work discussion.

---

> > ### Comment · Reviewer_ECDc · 2024-11-26
> >
> > Thank you for your response. It would be good to see more clear statement/explanation for Q2 and Q4 in the paper.

---

> > > ### Author Response · Authors · 2024-11-26
> > >
> > > **Q1. More clear statement/explanation for Q2 and Q4 in the paper.**
> > >
> > > Thank you for your suggestion. We have provided a clearer statement and explanation for Q2 in Appendix Section G (Resource Efficiency of the MOO Model) and for Q4 in Section 7 (Future Work) of the main paper.

---

### Author Response · Authors · 2024-11-24
**General response**

We sincerely appreciate the reviewers' thoughtful feedback and valuable suggestions. We have carefully addressed the concerns raised by the reviewers and made the following changes in the revised version of the paper:

- Added a new reference [a] in the **Related Work** section, as suggested by **Reviewer ZHdt (W4)**.
- Clearly outlined the differences between [b] and our method in **Section 3.2**, as suggested by **Reviewer KNnQ (Q6)**.
- Included new baseline results in **Tables 1, 2, and 5**, as recommended by **Reviewer KNnQ (Q4)** and **Reviewer ZHdt (Q1)**.
- Detailed hyperparameter settings for PT+FT methods in **Appendix E**, as suggested by **Reviewer KNnQ**.
- Provided a comprehensive description of baselines in **Appendix D**, as recommended by **Reviewer ZHdt (Q2)**.
- Added **Table 6** in the appendix, which includes a complete list of notations used in the paper.
- Made the results section more concise, refined the writing for clarity, and implemented minor changes as suggested by **Reviewer ZHdt (W4,5)** and **Reviewer ECDc (W3)**.

All changes are highlighted in blue for ease of reference in the main paper.



### Reference

[a] Radford et al., "Robust speech recognition via large-scale weak supervision," ICML, 2023.

[b] Saif et al., "Joint Unsupervised and Supervised Training for Automatic Speech Recognition via Bilevel Optimization," ICASSP 2024.

---

### Meta-Review · Area_Chair_H5B8 · 2024-12-18

**Metareview:**

In this paper, the authors studied multi-objective optimization (MOO) for multitask multilingual speech processing which has the issues of conflicting objectives during training. The authors proposed three multi-objective training objectives to mitigate the issues, and achieved better performance on automatic speech recognition (ASR) and speech-to-text translation (S2TT) across multiple languages. During rebuttal, the authors actively answered reviewers’ questions by providing additional details and clarifying misunderstandings.

The paper has novelties and should be able to provide valuable insights to audience who have interests in multitask multilingual speech processing. However, there are some issues with the paper.

The primary concern of this paper is the insufficient baseline. The authors, during the rebuttal, explained that they are utilizing a single model for multiple objectives, which accounts for the subpar performance compared to single-task models. Initially, the paper presented only two model configurations: Conformer 58M and 100M. In response to Reviewer ECDc's challenge, the authors included Table 8 with a model size of 300M parameters. This 300M model baseline significantly outperforms the proposed models with 58M and 100M sizes. This raises a question regarding multilingual modeling: why are the authors employing very small model sizes? Additionally, the proposed method shows less improvement with the 300M model compared to the smaller models (58M and 100M). Is it possible that the gains will diminish further with even larger models?

The complex objectives proposed in the paper are also not easy to scale up to even more languages. The paper only handles 5 languages.

In the community, Whisper or USM has demonstrated that it is possible to build a high-quality multilingual (around 100 languages) multitask model by using a simple objective function (without MOO) when increasing the model size.

The authors were challenged by reviewers why the results on Librispeech test other were not reported. After the authors provided the results in the rebuttal, it further concerned me. The gain of the proposed method is much less than the gain  on the easy test clean condition.

Another issue is that the paper’s focus is primarily on improving the performance of the ASR and S2TT tasks.  However, these two tasks are similar sequence classification tasks. It doesn’t bring too much challenge to optimization. However, tasks such as language identification, speaker verification, speech enhancement etc. are very different.  For example, ASR is to keep content information, disregarding speaker information, while speaker verification is the other way around. Therefore, the optimization objectives will be more conflicting to each other if ASR and speaker identification tasks are both included in training.

Overall, this paper is a decent work. However, it should be improved with stronger baseline and more obvious conflicting tasks to show the generalization of the proposed methods.

In summary, the strength of this paper is that the proposed three multi-objective training objectives are novel enough to mitigate the issues of conflicting objectives during training multitask multilingual models. The primary weakness of the paper is the insufficient baseline, raising questions about the actual effectiveness of the proposed method

**Additional Comments On Reviewer Discussion:**

There are a few key points raised by reviewers. The authors actively answered reviewers’ questions by providing additional details.  However, some details further reveal the weakness of the paper. Here are two examples.

1. The model size: Initially, the paper presented only two model configurations: Conformer 58M and 100M. In response to Reviewer ECDc's challenge, the authors included Table 8 with a model size of 300M parameters. This 300M model baseline significantly outperforms the proposed models with 58M and 100M sizes. This raises a question regarding multilingual modeling: why are the authors employing very small model sizes? Additionally, the proposed method shows less improvement with the 300M model compared to the smaller models (58M and 100M). Is it possible that the gains will diminish further with even larger models?

2. The Librispeech test other evaluation: Initially the authors didn't provide results. After being challenged by the reviewer, the authors provided results in the rebuttal and argued the reason of not providing results in the submission is to let the readers focus more on AIShell. This is not convincing. From the new results, it shows the proposed method has much less gain than the easy test clean condition.

Note that I ignored the review from Reviewer TuPN, which is not informative.

---

### Decision · Program_Chairs · 2025-01-22

Reject